# The development and diversity of religious cognition and behavior: Protocol for Wave 1 data collection with children and parents by the Developing Belief Network

Kara Weisman[1‡]*, Maliki E. Ghossainy[2‡]*, Allison J. Williams[2], Ayse Payir[2], Kirsten A. Lesage[2], Bolivar Reyes-Jaquez[1¤], Tamer G. Amin[3], Florencia K. Anggoro[4], Emily R. R. Burdett[5], Eva E. Chen[6], Lezanie Coetzee[7,8], John D. Coley[9], Audun Dahl[10], Jocelyn B. Dautel[11], Helen Elizabeth Davis[12,13], Elizabeth L. Davis[1], Gil Diesendruck[14], Denise Evans[7,8], Aidan Feeney[11], Michael Gurven[15], Benjamin D. Jee[16], Hannah J. Kramer[11], Tamar Kushnir[17], Natassa Kyriakopoulou[18], Katherine McAuliffe[19], Abby McLaughlin[19], Shaun Nichols[20], Ageliki Nicolopoulou[21], Peter C. Rockers[22], Laura Shneidman[23], Irini Skopeliti[24], Mahesh Srinivasan[25], Amanda R. Tarullo[26], Laura K. Taylor[27], Yue Yu[28], Meltem Yucel[17], Xin Zhao[29], Kathleen H. Corriveau[2‡]*, Rebekah A. Richert[1‡]*, on behalf of the Developing Belief Network[¶]

1 Department of Psychology, University of California Riverside, Riverside, California, United States of America, 2 Wheelock College of Education & Human Development, Boston University, Boston, Massachusetts, United States of America, 3 Department of Education, American University of Beirut, Beirut, Lebanon, 4 Department of Psychology, College of the Holy Cross, Worcester, Massachusetts, United States of America, 5 School of Psychology, University of Nottingham, Nottingham, United Kingdom, 6 College of Education, National Tsing Hua University, Hsinchu, Taiwan, R.O.C., 7 Health Economics and Epidemiology Research Office (HE2RO), T. H. Chan School of Public Health, Harvard University, Boston, Massachusetts, United States of America, 8 School of Clinical Medicine, Faculty of Health Sciences, University of the Witwatersrand, Johannesburg, South Africa, 9 Department of Psychology, College of Science, Northeastern University, Boston, Massachusetts, United States of America, 10 Department of Psychology, University of California Santa Cruz, Santa Cruz, California, United States of America, 11 School of Psychology, Queen's University Belfast, Belfast, United Kingdom, 12 School of Human Evolution and Social Change and the Institute of Human Origins, Arizona State University, Tempe, Arizona, United States of America, 13 Department of Human Evolutionary Biology, Harvard University, Cambridge, Massachusetts, United States of America, 14 Department of Psychology, Bar-Ilan University, Ramat-Gan, Israel, 15 Department of Anthropology, University of California Santa Barbara, Santa Barbara, California, United States of America, 16 Department of Psychology, Worcester State University, Worcester, Massachusetts, United States of America, 17 Department of Psychology and Neuroscience, Duke University, Durham, North Carolina, United States of America, 18 Department of Early Childhood Education, National and Kapodistrian University of Athens, Athens, Greece, 19 Department of Psychology and Neuroscience, Boston College, Chestnut Hill, Massachusetts, United States of America, 20 Sage School of Philosophy, Cornell University, Ithaca, New York, United States of America, 21 Department of Psychology, Lehigh University, Bethlehem, Pennsylvania, United States of America, 22 Department of Global Health, School of Public Health, Boston University, Boston, Massachusetts, United States of America, 23 Department of Psychology, Pacific Lutheran University, Tacoma, Washington, United States of America, 24 Department of Educational Science and Early Childhood Education, University of Patras, Patras, Greece, 25 Department of Psychology, University of California Berkeley, Berkeley, California, United States of America, 26 Department of Psychological and Brain Sciences, College of Arts and Sciences, Boston University, Boston, Massachusetts, United States of America, 27 School of Psychology, University College Dublin, Belfield, Dublin, Ireland, 28 Centre for Research in Child Development, National Institute of Education, Nanyang Technological University, Singapore, Singapore, 29 Department of Educational Psychology, East China Normal University, Shanghai, China

¤ Current address: Department of Psychology, University of New Hampshire, Durham, New Hampshire, United States of America

‡ KW and MEG are joint first authors on this work. KHC and RAR are joint senior authors on this work.

¶ Members of the Developing Belief Network (Wave 1) are listed in the Acknowledgments section; more information about membership is provided in the OSF repository associated with this manuscript (direct link to



**Data Availability Statement:** No datasets were generated or analysed during the current study. All

relevant data from this study will be made available upon study completion.

**Funding:** This work was supported by the John Templeton Foundation (https://www.templeton.org/) under grant #JTF61542, awarded to RAR and KHC. The funders did not and will not have a role in study design, data collection and analysis, decision to publish, or preparation of the manuscript.

**Competing interests:** The authors have declared that no competing interests exist.

membership information: https://osf.io/4gevr).

* kgweisman@gmail.com (KW); malikig@gmail.com (MEG); kcorriv@bu.edu (KHC); rebekah.richert@ucr.edu (RAR)

# Abstract

The Developing Belief Network is a consortium of researchers studying human development in diverse social-cultural settings, with a focus on the interplay between general cognitive development and culturally specific processes of socialization and cultural transmission in early and middle childhood. The current manuscript describes the study protocol for the network's first wave of data collection, which aims to explore the development and diversity of religious cognition and behavior. This work is guided by three key research questions: (1) How do children represent and reason about religious and supernatural agents? (2) How do children represent and reason about religion as an aspect of social identity? (3) How are religious and supernatural beliefs transmitted within and between generations? The protocol is designed to address these questions via a set of nine tasks for children between the ages of 4 and 10 years, a comprehensive survey completed by their parents/caregivers, and a task designed to elicit conversations between children and caregivers. This study is being conducted in 39 distinct cultural-religious groups (to date), spanning 17 countries and 13 languages. In this manuscript, we provide detailed descriptions of all elements of this study protocol, give a brief overview of the ways in which this protocol has been adapted for use in diverse religious communities, and present the final, English-language study materials for 6 of the 39 cultural-religious groups who are currently being recruited for this study: Protestant Americans, Catholic Americans, American members of the Church of Jesus Christ of Latter-day Saints, Jewish Americans, Muslim Americans, and religiously unaffiliated Americans.

## Introduction

In this paper, we describe a study protocol designed to examine the development and diversity of religious cognition and behavior across a wide range of cultural-religious settings. This protocol was designed collaboratively by a consortium of scholars (the Developing Belief Network) to be suitable for research with young children and their parents and caregivers in 39 distinct cultural-religious groups (to date).

Religion is a deeply important aspect of human experience, but the fields of psychology and developmental science have largely ignored and marginalized studies of religious beliefs and practices [1, 2]. Yet, as our team has argued elsewhere [3, 4], studies of religious beliefs and practices provide unique insights into the ways in which cognition and culture are mutually constituted [5, 6], and the ways in which children come to think, behave, and experience the world around them [7].

To give just a few examples, many religious and spiritual traditions hinge on concepts of disembodied spirits and deities, specify theories about what happens after death, prescribe rules about eating and clothing, and impart some sense that members of the faith share something important in common which differentiates them from others. Careful comparisons of such beliefs and practices across specific religious traditions have surfaced similarities that might be rooted in human cognitive tendencies shared across our species [8, 9]. At the same time, these comparisons often reveal important and sometimes dramatic differences, some of

which are at the root of many of the world's oldest, most violent, and seemingly most irreconcilable conflicts.

Because so much of religious cognition and behavior involves entities and phenomena that are understood to be difficult or impossible to observe directly (such as gods, souls, and supernatural forces [10]), learning about religion presents an unusual learning problem to the developing child [11, 12]. Unlike many of the conceptual domains that have been the focus of cognitive developmental research, such as intuitive physics or folk biology, learning about religious beliefs and behaviors is likely driven primarily, if not exclusively, by social processes: Instead of learning through direct observation or independent experimentation, children rely on processes such as cultural learning [e.g., 13], social learning [e.g., 14], explicit teaching [e.g., 15], learning through testimony [e.g., 12], learning through text [e.g., 16, 17], and learning through participation, observation, and pitching in [e.g., 18, 19]. In this sense, studies of the development of religious cognition and behavior provide unique insights into cognitive development and cultural transmission more broadly—processes of perennial interest to developmental scientists.

The aim of the current study is to draw on these insights to address this fundamental yet understudied aspect of human development and cognition.

We focus in particular on how children come to represent and reason about religious and supernatural agents, how children come to represent and reason about religion as an aspect of social identity, and the role that older children and caregivers play in transmitting religious beliefs within and between generations. In developing the current study protocol, we have adopted a philosophical approach, common in cognitive science and developmental psychology, that beliefs are attitudes held by individuals about what is true, which need not be explicit but can be expressed in language, and which play a causal role in behavior [20]. With regards to religion, people can have beliefs about the existence of entities (e.g., religious or supernatural beings and forces); beliefs about what defines members of one religious group and differentiates them from non-members; beliefs about the norms, rituals and practices of their religious community; and so on. Our primary goal in this study is to explore how these beliefs develop and change over early and middle childhood, and the similarities and differences in these developmental trajectories across diverse cultural-religious settings. Drawing on the cognitive, constructivist, and social constructivist traditions, we employ direct interviews with children to gain insights into their conceptual development, including their beliefs; as well as surveys of adult caregivers and observations of caregiver-child interactions to gain insights into one important aspect of the cultural transmission of these beliefs. (See [3] for an extended description of the broader aims, theoretical orientations, and methodological approach of the Developing Belief Network).

## Manuscript overview

The current manuscript introduces a study protocol designed to explore the development and diversity of religious cognition and behavior across diverse cultural-religious settings.

First, we introduce the Developing Belief Network: the consortium of developmental psychologists and other scholars that collaboratively designed the study protocol described here (see section: "The Developing Belief Network").

Next, we present the three questions that currently guide our research (see section: "Research questions").

We then turn to a comprehensive description of the study protocol (see section: "Materials and methods"). We begin by providing information about our sampling approach, an overview of the components of this protocol, and a description of our general approach to working

with the diverse cultural-religious groups included in this project. We then describe all components of the study protocol in full.

Taken as a whole, this study protocol is largely exploratory in nature; the primary goal is to describe the landscape of religious cognition and development within and across diverse settings. Nonetheless, most components of the protocol have been designed with certain analyses in mind, which we describe in "Planned Analyses" sections at the end of each task description. We consider these to be the most basic analyses of each task, which will lay the foundation for a wide variety of additional analysis approaches, both by current members of the Developing Belief Network and by future researchers.

For each component of the protocol, we also provide sample-specific details for the six cultural-religious groups being recruited in the United States: Protestant Americans, Catholic Americans, American members of the Church of Jesus Christ of Latter-day Saints, Jewish Americans, Muslim Americans, and religiously unaffiliated Americans living in and around Riverside, California, or Boston, Massachusetts. These are locations in which the lead and senior authors have extensive cultural expertise and past experience conducting research on religious development. Full English-language protocols for these samples are available in the following OSF repository: https://osf.io/dumf4/. Future manuscripts will present sample-specific adaptations for field sites and samples outside of the US and a full description of the extensive, iterative processes of translation and cultural adaptation underlying the implementation of this protocol in the additional 33 cultural-religious groups included in this study (to date). Here, we note that the translation process has varied across research teams and field sites/samples, depending on the nature of the languages in question, the linguistic expertise of the research team, the budget allocated for translation, and typical workflow of the research team in previous projects in their field site(s). In contrast, cultural adaptation—that is, the selection of sample-specific stimuli and the practical implementation of these methods in a particular site and sample—has been much more centralized, involving extensive back-and-forth between research team leaders and the DBN's "core leadership team" (PIs: Rebekah Richert, Kathleen Corriveau; additional leadership team members: Maliki Ghossainy, Kirsten Lesage, Ayse Payir, Bolivar Reyes-Jaquez, Kara Weisman, Allison Williams) to establish, revise, and uphold network-wide standards and procedures.

We conclude the current manuscript with guidance to future researchers hoping to use the datasets that will emerge from this protocol and to those seeking to adapt and extend this protocol to new field sites and samples (see section: "Discussion").

## The Developing Belief Network

The Developing Belief Network (DBN) is a consortium of researchers studying human development in diverse social-cultural settings, with a focus on the interplay between general cognitive development and culturally specific processes of socialization and cultural transmission in early and middle childhood. PIs Rebekah Richert and Kathleen Corriveau launched the DBN in 2020 through funding from the John Templeton Foundation. The DBN currently consists of a core leadership team based jointly at the PIs' home institutions (the University of California, Riverside, and Boston University), and an additional eight research teams selected through a competitive application process.

The DBN's first wave of data collection aims to explore the development and diversity of religious cognition and behavior, with a particular focus on the three research questions described in the next section: (1) How do children represent and reason about religious and supernatural agents? (2) How do children represent and reason about religion as an aspect of social identity? (3) How are religious and supernatural beliefs transmitted within and between generations?

This paper describes the study protocol designed to address these research questions. This protocol was designed over the course of roughly 18 months through an iterative process of consultation with community members and religious experts at research sites, discussion across research teams, pilot testing, revision, and network-wide consensus building.

This research is currently taking place in 39 distinct cultural-religious settings, spanning 17 countries and at least 13 languages. We list these field sites and samples here by broad geographical regions to give a sense of the wide range of cultural and religious groups represented in this data collection effort; see also Fig 1.

In the Americas, the study is being administered in English (and in Spanish as needed) to participants identified as Protestant, Catholic, members of the Church of Jesus Christ of Latter-day Saints, Jewish, Muslim, or religiously unaffiliated living in and around Riverside, California, United States, and Boston, Massachusetts, United States (PIs: Rebekah Richert and Kathleen Corriveau); in Yucatec Maya and Spanish to Catholics and a diverse group of Evangelical, Pentecostal, and Anabaptist Protestant Christians living in Maya villages in the Yucatán, Mexico (PI: Rebekah Richert; co-PIs: Laura Shneidman and Elizabeth Davis); in Spanish to Catholics and Evangelical Protestants living in the Mantaro Valley, Peru (PI: Katherine McAuliffe); and in Tsimané and Spanish to Evangelical Protestants living in Tsimané villages in the Bolivian Amazon (PI: Emily Burdett; co-PIs: Helen Elizabeth Davis and Michael Gurven).

In Europe, the study is being administered in English to Catholics and Protestants living in Northern Ireland (PI: Jocelyn Dautel; co-PIs: Laura Taylor, Aidan Feeney, and John Coley); in English to Catholics and Protestants living in the Republic of Ireland (PI: Jocelyn Dautel; co-

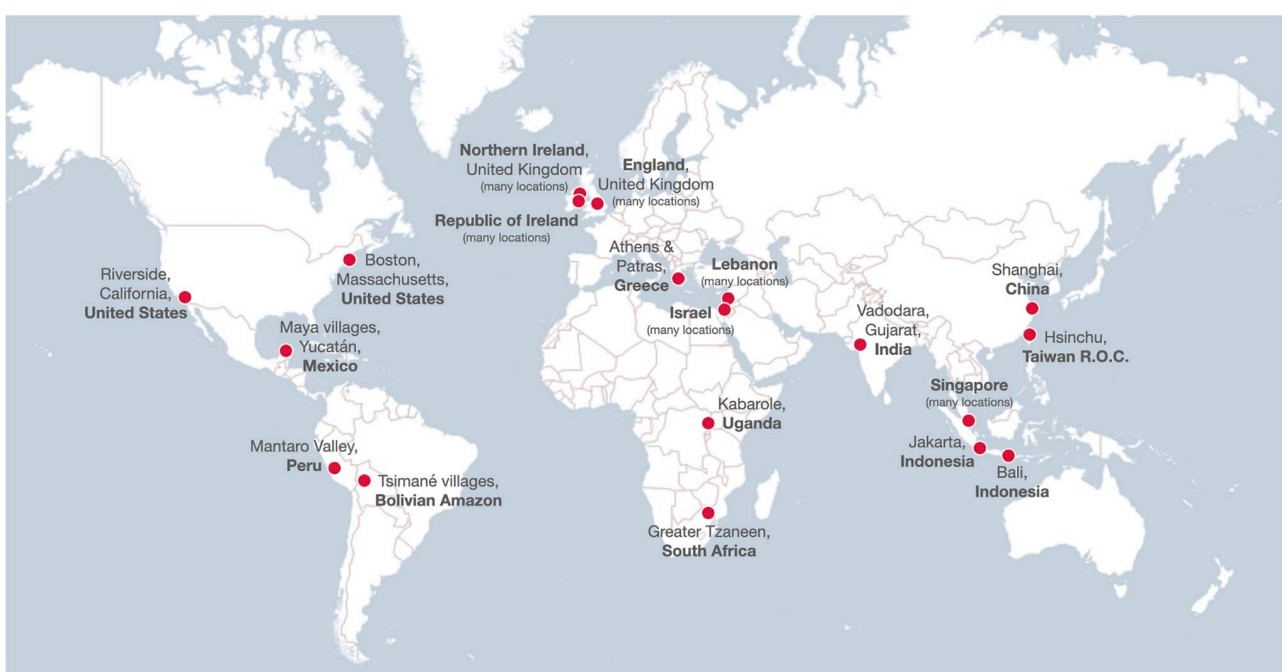

**Fig 1. Locations for data collection.** For an interactive version of this map, see http://u.osmfr.org/m/830909/. Within each location, this research is being conducted with samples from 1–6 distinct cultural-religious groups, for a total of 39 cultural-religious groups spanning 17 countries and at least 13 languages. This map was created using the open source mapping software uMap (an OpenStreetMap project: https://umap.openstreetmap.fr/en/). OpenStreetMap® is open data (see https://www.openstreetmap.org/copyright), licensed under the Open Data Commons Open Database License (ODbL).

PIs: Laura Taylor, Aidan Feeney, and John Coley); in English to Christians and Muslims living in England (PI: Emily Burdett; co-PIs: Helen Elizabeth Davis and Michael Gurven); and in Greek to Greek Orthodox Christians living in and around Athens and Patras, Greece (PIs: Kathleen Corriveau and Rebekah Richert; co-PIs: Ageliki Nicolopoulou, Natassa Kyriakopoulou, and Irini Skopeliti).

In the Middle East, the study is being administered in Arabic (with limited code-switching to French and English as needed) to Maronite Catholics, Orthodox Christians, Protestant Christians, Shia Muslims, and Sunni Muslims living throughout Lebanon, as well as members of the Druze faith living in and around the Chouf Mountains, Lebanon (PI: Tamer Amin; co-PI: Maliki E. Ghossainy); in Hebrew to Modern Orthodox Jewish people living in and around Tel Aviv, Israel (PI: Mahesh Srinivasan; co-PIs: Gil Diesendruck and Audun Dahl); and in Arabic to Muslim Arab people living in Central and Northern Israel (PI: Mahesh Srinivasan; co-PIs: Gil Diesendruck and Audun Dahl).

In sub-Saharan Africa, the study is being administered in Rutooro to Anglicans and Catholics living in and around Kabarole, Uganda (PI: Katherine McAuliffe); and in Xitsonga and Sepedi to Christians with a range of syncretic traditional Southern African beliefs and practices living in the rural villages surrounding Tzaneen, South Africa (PI: Amanda Tarullo; co-PIs: Denise Evans and Peter Rockers).

In South Asia, the study is being administered in Hindi to Hindus and Muslims living in Vadodara, Gujarat, India (PI: Mahesh Srinivasan; co-PIs: Audun Dahl and Gil Diesendruck).

In Southeast Asia, the study is being administered in English to Buddhists and Muslims living in Singapore (PI: Tamar Kushnir; co-PIs: Yue Yu and Xin [Alice] Zhao); and in Indonesian to Muslims and Christians (Protestants and Catholics) living in Jakarta, Indonesia, and to Hindus and Muslims living in Bali, Indonesia (PI: Florencia Anggoro; co-PI: Benjamin Jee).

Finally, in East Asia, the study is being administered in Mandarin Chinese (written in Simplified Chinese) to religiously unaffiliated people living in and around Shanghai, China (PI: Tamar Kushnir; co-PIs: Xin [Alice] Zhao and Yue Yu); and in Mandarin Chinese (written in Traditional Chinese) to people who engage in a range of Buddhist, Taoist, and Yiguandao beliefs and practices living in and around Hsinchu, Taiwan R.O.C. (PI: Kathleen Corriveau; co-PI: Eva Chen).

The DBN is planning for three waves of data collection, following the same samples of children over the course of three years. This manuscript focuses on the methods for Wave 1, which will shed light on development via cross-sectional comparisons of ages. Future waves of data collection will complement this cross-sectional approach with longitudinal analyses of individual children.

As of the initial submission of this manuscript (December 2022), Wave 1 data collection had begun with 27 (68%) of these samples, including five samples in which data collection has been completed (the two samples collected in Peru, the two samples collected in Uganda, and one of the four samples being collected in Indonesia). As of August 2023, Wave 1 data collection has begun in 97% of samples.

## Research questions

The inaugural members of the DBN undertook an interactive, collaborative process of drafting and revising the research questions that have guided study design for Wave 1. This initial round of data collection prioritizes the exploration of two broad topics of interest: the diversity of religious and supernatural concepts that children encounter, and the ways in which children come to understand and construct religiously-based social categories. For both topics, our goals are to uncover and describe children's representations and reasoning, to chart age-

related changes across early and middle childhood, to assess similarities and differences across cultural and religious settings, and to gain insight into the processes of social-cultural transmission that scaffolds children's learning about how to think and act like the adults in their communities. The study covers a wide age range, to chart the developmental progression of religious cognition as well as to understand how children at their developmental level grapple with religion and the supernatural. These goals are articulated in three research questions detailed below.

**Research question 1: How do children represent and reason about religious and supernatural agents?.** Our first research question focuses on the religious and supernatural agents that play a role in most world religions and in many traditional spiritual beliefs and practices, such as gods, spirits, and ancestors.

The study of "agents" in general (i.e., people and other beings with experiences, intentions, or the ability to form relationships) has been a core focus of developmental psychology for many decades [e.g., 21–30]; and religious and supernatural agents in particular have received widespread attention in the cognitive science of religion [e.g., 8, 31–39]. Our goal is to add to the growing number of studies attempting to bridge these two fields.

Within this broad research question, the DBN has several sub-goals. First, we aim to describe the range of religious and supernatural agents with which children in a given setting are familiar, and the extent to which these agents are considered "real." Our second goal is to assess which of these agents are the most relevant to children's lives, and in what ways. For example, do parents identify certain agents as especially critical for children to learn about? Are certain agents salient in holidays or rituals? Third, we aim to explore how "folk theories" (e.g., folk physics, biology, and psychology) support and constrain concepts of and beliefs in religious agents, and, conversely, how exposure to religious and supernatural beliefs influences the development of folk theories. Finally, we aim to assess how individual differences in domain-general cognitive skills and capacities (e.g., aspects of executive function) relate to children's developing conceptualizations of religious and supernatural agents.

**Research question 2: How do children represent and reason about religion as an aspect of social identity?.** Our second research question concerns how children come to understand and construct social categories based on religion or ethno-religious group membership. How do children become aware of the social meaning of religion and religious affiliation, both for themselves and for the people around them?

When raised in a particular belief system, religious beliefs are inextricable aspects of identity. Many people around the world perceive themselves and others, and make sense of the world, through the lens their religion provides; thus, in addition to being a fascinating topic in its own right, the formation of religious and ethno-religious identity has important implications for a number of pressing social issues, such as immigration, climate change, gender identity, and minority rights. At the same time, the development of concepts and categories related to religious and ethno-religious identity has received less attention in the burgeoning field of social cognitive development than categories like gender, race, and language, which have been the focus of many studies in the past 25 years [e.g., 40–46]. Our goal in the current research is to add to the small but growing number of studies attempting to integrate religious identity into our understanding of the development of social categories in early and middle childhood.

Within this broad research question, the DBN has several sub-goals. First, we aim to assess when and on what basis children come to self-identify as a member of a particular religious group. Second, we aim to chart children's developing awareness of the various indicators of religious identity and religiosity (e.g., how to tell if someone is a member of one's own religious group), as well as the degree to which children essentialize religious groups. Related to both of these sub-goals, our third aim is to explore how the conceptualization and development of

religious identity relate to other aspects of the social-cultural setting, such as the degree of religiosity in the broader community, or children's exposure to religious diversity or religious conflict. Finally, our fourth aim is to explore how children's developing sense of religious identity shapes their understanding of the various social norms (religious, moral, conventional, and otherwise) that guide people's behaviors.

**Research question 3: How are religious and supernatural beliefs transmitted within and between generations?.** Our third research question concerns the processes of social-cultural transmission that scaffold children's learning about how to think and act like the adults in their communities. How do children weigh different sources of information in forming their own religious beliefs and identities? How do parents and other caregivers conceptualize and participate in the processes of socialization and enculturation that encourage children to become "good" members of their local faith communities and their broader social worlds?

As described earlier, reasoning about religious phenomena poses a unique challenge to young children, as these phenomena are largely understood to be unobservable. If children cannot learn about god(s), the afterlife, or the mechanisms of prayer via firsthand experience, they must rely primarily (if not exclusively) on other, more "social" forms of learning [e.g., 12–19]. Studying how caregivers conceptualize and participate in their children's religious and spiritual development enables us to explore this unique learning process, and to document similarities and differences across cultures in the transmission of beliefs about unobservable phenomena.

The current protocol applies this broad research question both to the development of beliefs about religious and supernatural agents (Research Question 1) and the development of religious identity (Research Question 2). For example, regarding religious and supernatural agents: Which agents do children learn about from specific sources (e.g., parents, peers, formal schooling, or religious leaders), and in what ways (e.g., through conversation, instruction, observation, ritual, stories, and so on)? Regarding religious identity: How do parents and other caregivers discuss religious or other social conflicts with children, and how is this modulated by the family's exposure to religious diversity or religious conflict in their broader community?

Within this research question, we have several sub-goals. First, we aim to assess how children weigh different sources of information, and how such relative weighting of these information sources might change over development. Second, we aim to document parents' cultural belief systems ("ethnotheories") about when, how, and from which sources children *should* learn (e.g., their views on what are or are not appropriate pedagogical practices or sources). Third, we have a general interest in documenting and describing the religious and supernatural rituals, ceremonies, holidays, and artifacts that are most salient and relevant to children in each setting, considering both children's home lives, and their participation in places and events that are explicitly marked as "religious," with an eye toward exploring how these aspects of the social-cultural environment shape children's developing beliefs and behaviors. By using open-ended questions to probe the sources of children's religious cognition and identity, the tasks that address Research Question 3 can inform future studies of specific contexts in which children learn about religion, for instance by naturalistic observations of parent-child interactions, peer conversations, or school contexts.

## Materials and methods

The current study protocol is designed to address the research questions described above via three protocol elements: (1) the Child Protocol, a set of behavioral tasks for children between the ages of 4–10 years; (2) the Parent Survey, a comprehensive survey to be completed by children's parents or other primary caregivers; and (3) a Caregiver-Child Conversation task.

As described above, this study is currently being conducted in 39 distinct cultural-religious settings (to date), spanning 17 countries and 13 languages. All research teams aim to collect at least 40 children and their caregivers for each cultural-religious sample. Beyond this, planned sample size, eligibility, inclusion and exclusion criteria, recruitment strategies, the mode of data collection, and compensation vary across samples; these decisions are made by field site leaders in close consultation with the core leadership team. Sample-specific item selection also varies across samples, within the network-wide guidelines for each task that are detailed in the current manuscript. In addition to establishing these guidelines, this manuscript provides a detailed description of the adaptation and implementation of this protocol in the US, specifically within six cultural-religious samples: American Protestants, American Catholics, American members of the Church of Jesus Christ of Latter-day Saints, Jewish Americans, Muslim Americans, and religiously unaffiliated Americans. Future manuscripts will provide details of sample-specific design and sampling approaches for the other field sites and samples included in this research.

For the full text of all materials, all visual stimuli, and Qualtrics survey (.qsf) files, please visit https://osf.io/dumf4/.

## Samples

**Sample size (US samples).** For each of the six US samples, we have set a planned minimum sample of 75 children (and their parents/caregivers) and a planned maximum sample of 100 children (and their parents/caregivers). The planned minimum of 75 reflects the largest minimum per sample we could imagine attaining given the US data collection staff and budget and the timing of this first wave of data collection (part of a three-wave longitudinal study, funded by a grant with an end date approximately 3 years after data collection began); the planned maximum of 100 reflects our estimation of a reasonable target goal given these practical constraints. This yields a total planned sample of 450 to 600 children. For each sample, our original plan was to stop collecting data when we have reached our planned maximum sample size, or on September 30, 2023, whichever came first; over the course of revising the current manuscript, we have encountered substantial difficulties recruiting participants from minoritized religious groups in the US (i.e., our Church of Jesus Christ of Latter-day Saints, Jewish, and Muslim samples), and have decided to extend the stop date to December 31, 2023. A sample is considered "complete" if it meets or exceeds the minimum sample size; if we do not meet that sample size by our end date, then any participants from the relevant group will only be included in analyses that pool across multiple samples.

(Please note that sampling plans vary substantially across the rest of the cultural-religious samples represented in the network, and will be provided in forthcoming manuscripts.)

**Eligibility, inclusion, and exclusion criteria (US samples).** US samples are being recruited through the University of California, Riverside, and Boston University, using a shared set of eligibility, inclusion, and exclusion criteria.

To be eligible for inclusion, a family must currently live within driving distance of one of the two university campuses. Although data collection for Wave 1 in the US is remote due to the ongoing COVID-19 pandemic (see "Recruitment, data collection, and compensation [US samples]," below), this geographical restriction is intended to allow for in-person participation for future waves of data collection. Participating children must be between the ages of 4 years 0 months and 10 years 11.99 months at the time of the initial study session. Children must speak English or Spanish, and parents/caregivers must read and respond in writing in English or Spanish. Participating parents/caregivers must be parents or legal guardians of the participating child; multiple siblings from the same family are allowed to participate. Finally, parents/

caregivers must report that children are either (a) being raised in one of the following religious traditions: Protestant Christianity, Catholicism, the Church of Jesus Christ of Latter-day Saints, Islam, or Judaism; or (b) being raised in no religious tradition (i.e., the child is religiously unaffiliated). Children who are being raised in other religious traditions (e.g., Buddhism, Hinduism, Orthodox Christianity, or Sikhism) are ineligible for the study; in these cases, families are thanked for their interest, and invited to participate in other studies in the relevant lab.

To determine which version of the study is most appropriate for each participating child, we ask parents/caregivers interested in participating in the study to complete an initial "Family Interest Form," in which they respond to the following questions about their family's religion: (1) "What is your religion, if any? Please be specific here by including any denomination(s), branch(es), or type(s) that you belong to, for example: Christian—Lutheran; Roman Catholic; Sunni Muslim; Shia Muslim; Orthodox Jew; Therevada Buddhist; Mahayana Buddhist; Not Religious—Agnostic."; (2) "Does your child have the same religion (if any) as you?"; and, if the answer to the second question is no, (3) "What is your child's religion, if any? Please be specific here by including any denomination(s), branch(es), or type(s) that your child belongs to, for example: Christian—Lutheran; Roman Catholic; Sunni Muslim; Shia Muslim; Orthodox Jew; Therevada Buddhist; Mahayana Buddhist; Not Religious—Agnostic"; and (4) "Which of the following religious and cultural traditions is your child most familiar with? You might take into account the religious background of other members of your extended family, or the religious traditions that are most common in your child's neighborhood or school if your child is not religious."

All religious children are given the version of the protocol that corresponds to the religion in which the parent indicates the child is being raised. Note that a child's religious affiliation is assessed independently of their parents/caregivers' religious affiliation(s). For example, if a parent indicates that he is Muslim (question 1), that the child does *not* have the same religion as him (question 2), and that the child is Protestant (question 3), then we consider this child Protestant for sampling purposes and the child is shown the Protestant version of the protocol.

When a parent/caregiver indicates that a child is being raised in *multiple* religious traditions, the child is shown the protocol corresponding to the religion with which the parent/caregiver says the child is most familiar, and these children are included in the sample for that religious group. Roughly halfway through data collection, we will assess how many mixed-faith children have been included in each cultural-religious group, and whether their responses diverge substantially from other participants. If so, we plan to replace these participants with new participants from families who report the child being raised solely or primarily in that religious tradition.

When a parent/caregiver indicates a child is religiously unaffiliated, the child is given the protocol version associated with the religious tradition most familiar to the child. If the parent/caregiver says that the child is not familiar with any religion and there are no religions mentioned in any other part of the Family Interest Form, the child is given the Protestant protocol; this reflects our understanding of Protestant Christianity as a pervasive cultural-religious force across US history, and the most common religious affiliation in the US through the present day [47].

In all cases–for religious children and for unaffiliated children–the parent/caregiver is given the version of the Parent Survey that matches the version of the protocol that the child has been given, regardless of the parent/caregiver's own religious affiliation and familiarity with religion. We tell families that if there is one parent/caregiver whose religious affiliation (or familiarity with a specific religion) is more similar to the child's, we would prefer that individual to complete the Parent Survey and Caregiver-Child Conversation.

The research team working in the US has adopted two exclusion criteria. First, children who do not appear to understand the majority of the Child Protocol, whose parents/caregivers frequently provide translations to the child in a language other than English or Spanish, or whose parents/caregivers spontaneously self-report concerns about their child's comprehension, verbal expression, working memory, or other factors are excluded from the sample. Second, children whose sessions are frequently interrupted (e.g., by family members making suggestions about how the child should answer) are excluded from the sample; judgments of the frequency and severity of these interruptions are made by data collectors and discussed with other data collectors and supervisors from both universities (Boston University and the University of California, Riverside). We have no current plans to exclude data at the trial or task level, i.e., if a child's data is included in the dataset our default will be to include all of their data. However, there might be some exceptions to this made on a case-by-case basis during regular meetings of the data collection team. All decisions regarding exclusion are made as a team with data collectors from both universities and at least one supervisor.

(Please note that eligibility, inclusion, and exclusion criteria vary substantially across the rest of the cultural-religious samples represented in the network, and will be provided in forthcoming manuscripts.)

**Recruitment, data collection, and compensation (US samples).** Participants are recruited through a combination of participant databases housed at the University of California, Riverside, and Boston University; targeted Facebook ads; community outreach; and personal outreach to religious leaders and organizations in the local areas surrounding the two campuses.

Given the ongoing COVID-19 pandemic and the high levels of access and exposure to computer technology, data collection in the US is being conducted entirely online. The Child Protocol is administered via live video chat between a researcher and the participating child. The session is recorded via Zoom (except in cases where the parent/caregiver or child declines to be recorded), and the researcher also logs most of the child's responses in real-time via Qualtrics survey software. In some cases, the Caregiver-Child Conversation task is completed independently and asynchronously by parents/caregivers and children without live supervision by a researcher, and parents/caregivers submit an audio recording of this conversation to the research team via email. In other cases, the researcher administers the task to parents/caregivers and children at the end of the Child Protocol, or in a separate live video chat. Finally, parents/caregivers complete an online version of the Parent Survey independently and asynchronously via Qualtrics.

To thank participants, families are given one $20 e-gift card for each participating child upon completion of the Child Protocol, and an additional $20 gift card when the parent/caregiver completes the Caregiver-Child Conversation task and the Parent Survey. For families with more than one child participating, parents/caregivers receive an additional $10 gift card for completing the Caregiver-Child Conversation and Parent Survey with each additional child.

(Please note that recruitment, data collection procedures, and compensation vary substantially across the rest of the cultural-religious samples represented in the network, and will be provided in forthcoming manuscripts.)

## Overview

**Protocol components.** The study consists of three components: (1) the Child Protocol, (2) the Caregiver-Child Conversation task, and (3) the Parent Survey; see Fig 2 for overview. For the full text of all materials (as adapted for the five target religions included in US samples),

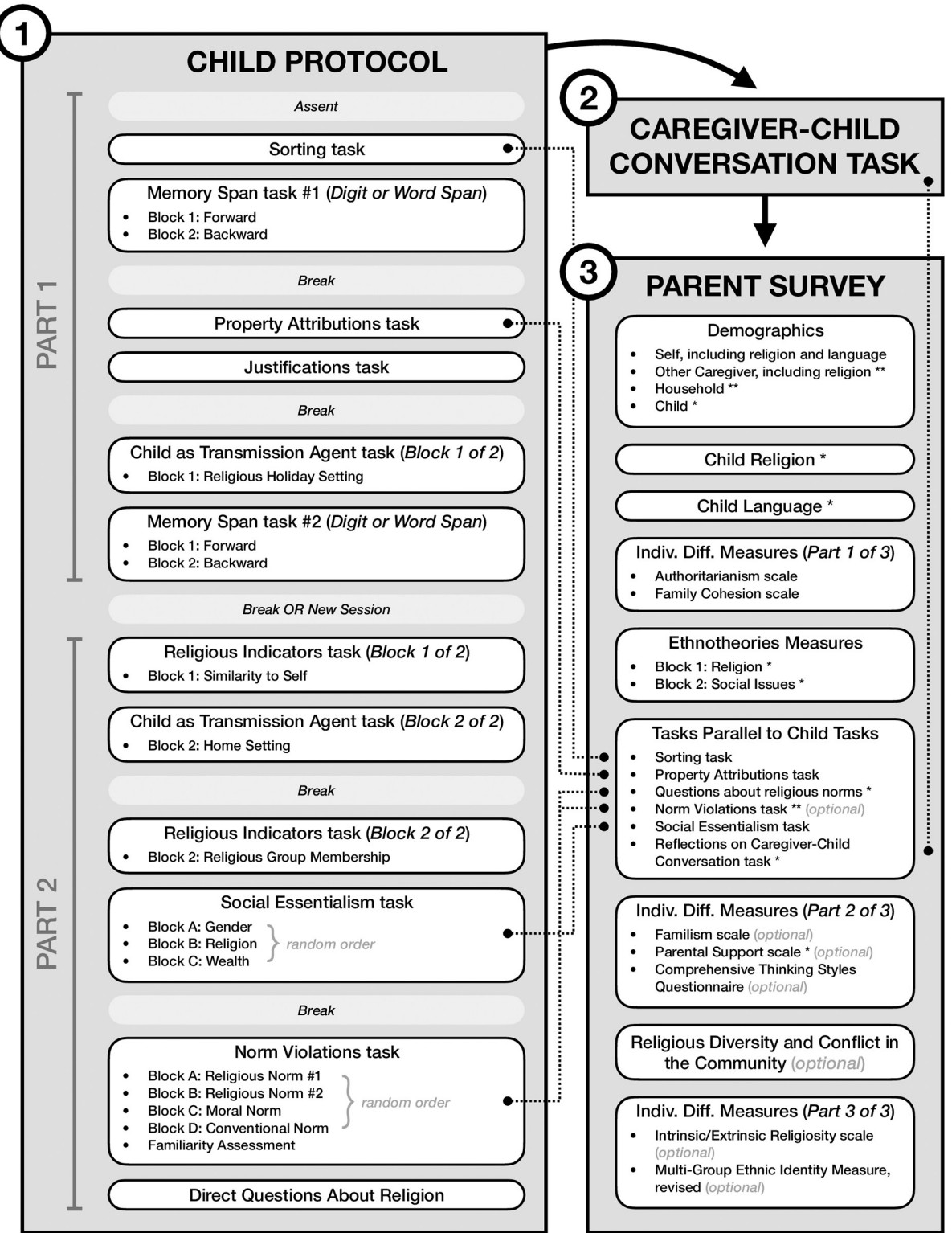

**Fig 2. Overview of the full Developing Belief Network Wave 1 study protocol for children (ages 4–10 years) and their parents or other primary caregivers.** Depicted here is the default order of study protocol components. First, the participating child completes the Child Protocol. Parts 1 and 2 of the Child Protocol are administered in the same testing session for some children (including most US children 8 years old or older), and in two testing sessions on different days for other children (including all US children under the age of 8 years). The order of the two Memory Span tasks (digit span and word span) is determined randomly for each child. Within each task, blocks labeled with

numbers (i.e., "Block 1, Block 2,. . .") are presented in the fixed order shown here, and blocks labeled with letters (i.e., "Block A, Block B,. . .") are presented in a random order for each child. Second, the child and their parent/caregiver participate in a Caregiver-Child Conversation task. In some cases, this task is completed on the same day as the Child Protocol, and in other cases it is completed on a different day. Third, the child's parent/caregiver completes the Parent Survey. Different samples of parents/caregivers are shown different subsets of the tasks marked as "optional"; US caregivers complete all of the tasks shown here. Any parents/caregivers who have more than one child participating in the study complete an abbreviated Parent Survey for the second (third, etc.) child, consisting of only the survey elements marked with one asterisk (*) as well as shortened or otherwise modified versions of the survey elements marked with two asterisks (**). "Indiv. Diff." = Individual Difference.

all visual stimuli, and Qualtrics survey (.qsf) files, please visit the following OSF repository: https://osf.io/dumf4/.

**Sample-specific adaptations.**   A primary goal of this project is to understand children's views around the kinds of religious entities, practices, and norms they encounter in their everyday lives [48]. To this end, for each sample we tailor the Child Protocol and Parent Survey to include stimuli that are salient to the particular cultural-religious setting in question, while maintaining a high degree of standardization in the structure and framing of all protocol elements. To make these sample-specific adaptations, each research team conducted semi-structured interviews with local religious experts relevant to their planned samples. As a network, we engaged in an iterative process of selecting stimuli, comparing and discussing choices across research teams, pilot testing, revising network-wide guidelines for stimulus selection, and revising sample-specific stimuli. The details of this process will be discussed in a separate, forthcoming manuscript.

In the current manuscript we present the standard versions and network-guidelines for each element of the study protocol, organized with respect to our three primary research questions. For each protocol element, we also detail the sample-specific adaptations for the five target religions included in US samples, where sample-specific adaptation is complete and data collection is underway.

### Research question 1: How do children represent and reason about religious and supernatural agents?

**Sorting task: Assessing the familiarity and reality status of religious agents.**   Building on previous work on children's understanding of what is real vs. not real [e.g., 49] as well as children's ability to sort items based on some type of rule [i.e., familiarity and reality status; see, e.g., 50], the Sorting task is designed to assess children's familiarity with natural, supernatural, and religious agents, and their judgments about which of these agents are real. This task is part of the Child Protocol; see Fig 2.

*Task design and procedure.* To begin the task, the researcher says, "We're going to talk about a bunch of different things. Your first job is to tell me if you've ever heard of the thing I mention. Then, your second job is to tell me if you think the thing is real or not real. Let's try some."

Next, the child is presented with two practice trials in a random order. One practice trial involves a familiar agent ("doctors"), and the second practice trial involves the use of a nonsense word that children should indicate is unfamiliar ("breksters"). These practice trials are designed to familiarize children with the task format; to encourage them to use the full range of response options available to them; to encourage them to take the task seriously (i.e., provide sincere responses even if the questions seem silly or unusual); and to establish that the task will include both familiar and unfamiliar and both real and unreal agents.

Immediately following the practice trials, the child is presented with roughly 15–20 test trials in a random order. On each trial, the child is asked two questions about an agent, in a fixed order: "Have you ever heard of [agent]?" (familiarity judgment; response options: "yes" or "no") and "Is/are [agent] real or not real?" (reality judgment; response options: "real" or "not real").

The range of agents included in this task is intended to vary across samples to preserve cultural and religious specificity and appropriateness, within certain limits designed to maintain a balanced level of standardization across samples. The aspects that were standardized are described in the following paragraphs; see Table 1 for items selected for each US sample.

Each version of the protocol contains at least six *religious or supernatural agents* that are widely considered "real" in the cultural-religious setting. These agents include the "Biggest" god [51] in the cultural-religious setting, as well as 1–2 additional prominent religious agents; these agents are also featured later in the protocol in the Property Attributions and

**Table 1. Sample-specific item selection for the Sorting task, for the five religions included in US samples.**

| Religion | | | | |
|---|---|---|---|---|
| *Protestantism* | *Catholicism* | *Church of Jesus Christ of L.D.S.* | *Judaism* | *Islam* |
| *Religious and supernatural agents* | | | | |
| God | God | God | God[a] | Allah |
| Jesus | Jesus | Jesus | Elijah | the Prophet Muhammad |
| the Holy Spirit | the Holy Spirit | the Holy Spirit | Jesus | Jesus |
| angels | angels | angels | angels | angels |
| the Devil | the Devil | the Devil | the Devil | the Devil |
| ghosts | ghosts | ghosts | ghosts | ghosts |
| fortune tellers | fortune tellers | fortune tellers | fortune tellers | fortune tellers |
| *Fictional agents* | | | | |
| fairies | fairies | fairies | fairies | fairies |
| witches | witches | witches | witches | witches |
| Santa Claus | Santa Claus | Santa Claus | Santa Claus | Santa Claus |
| zombies | zombies | zombies | zombies | zombies |
| *Natural and scientific agents* | | | | |
| germs | germs | germs | germs | germs |
| robbers | robbers | robbers | robbers | robbers |
| *Religious groups* | | | | |
| Christian people | Catholic people | members of the Church of Jesus Christ | Jewish people | Muslim people |
| Muslim people | Muslim people | Muslim people | Christian people | Christian people |
| *Attention/comprehension checks* | | | | |
| Blambies | Blambies | Blambies | Blambies | Blambies |
| the Rumber | the Rumber | the Rumber | the Rumber | the Rumber |
| *Sample-specific additions* | | | | |
| praying to God | praying to God | praying to God | praying to God | praying to God |
| – | – | prophets | prophets | prophets |
| – | – | – | – | djinn |

Unaffiliated children in the US are shown the version of the protocol that corresponds to the religion with which they are likely most familiar (see main text for details). See main text for descriptions of network-wide guidelines for item selection. "L.D.S." stands for "Latter-day Saints."

[a] If the researcher becomes aware that the child is more familiar with the term "Hashem" or "Adonai" then the more familiar term is used in place of "God" throughout the Child Protocol.

Justifications tasks (described in later sections). Beyond this, other religious/supernatural agents include a positive divine being; a negative divine being; dead humans that may or may not be worshiped; and humans with supernatural powers which at least some adults might believe are real.

Each version of the protocol also includes at least three *fictional agents* that are widely considered "fictional" or "fantasy" in the cultural-religious setting. These agents include one human with supernatural powers and two additional agents that are part of local folklore, including at least one agent with fairly "human-like" properties.

To provide a contrast to the religious, supernatural, and fictional agents described above, we include at least two *natural or scientific agents*: scientific entities unobservable by naked eye; and ordinary (non-supernatural) humans engaged in a negatively valenced or antisocial activity.

Each version of the protocol includes at least two *religious groups*, including the child's own religious group or a locally prevalent religious group, as well as at least one other religious group salient in the local setting. These two religious groups are also featured later in the protocol in the Religious Indicators, Social Essentialism, Norm Violations, and Child as Transmission Agent tasks (described in later sections). For these two trials, the child is asked one additional follow-up question after the familiarity and reality status questions. If the child indicates that they are familiar with the group in question, the follow-up question is, "What do you know about [religious group]?" If the child indicates that they are not familiar with the group in question, the follow-up question is instead, "When I say the word(s) [religious group], what do you think of?"

Finally, each version of the protocol includes two agents intended as *attention/comprehension checks*: a novel category of entities presented in the plural generic (e.g., "Blambies"); and a novel entity presented as a singular individual (e.g., "The Rumber").

All agents are presented in random order.

Research teams are encouraged to include a mix of specific individuals (e.g., "Santa Claus") and plural generics (e.g., "zombies"), as relevant in the local language and cultural-religious setting. We do not specify which items should be worded in which way; in other words, we prioritize cultural specificity within a sample over syntactical standardization across samples.

Some research teams also include additional, sample-specific items in this list as desired, in order to capture aspects of the cultural-religious setting beyond what is captured by the standard list of agents. Additional items are randomized along with the other agents in the standard list, with the exception of any additional ethno-religious groups beyond the two described above, which are presented in a random order after the standard list.

In the US samples, this task is estimated to take 5 to 7 minutes with younger participants and 4 to 9 minutes with the oldest participants.

*Adaptation for US samples*. The final set of agents featured in the Sorting task for the five religions included in US samples is presented in Table 1.

*Parallel task for parents/caregivers*. Parents/caregivers complete a modified version of the Sorting task just described. Each parent/caregiver is presented with the same set of agents as their child assessed, in a random order. For each agent, parents/caregivers are asked two questions: "Is/are [agent] real or not real?" (response options: "real," "not real," "I have never heard of this") and "How sure are you that [agent] is/are [real/not real]" (11-point response scale ranging from [0] "I am not sure at all" to [10] "I am extremely sure").

*Planned analyses*. We hypothesize that children's familiarity with the agents included in this task will increase with age, and that this increase will be more pronounced for supernatural agents [see 49, 50, 52, 53]. To test this, we will examine the main effect of participant age (4–10 years) on responses to the familiarity question ("Have you heard of [agent]?"), as well as

statistical interactions between participant age and agent type (supernatural vs. natural, fictional vs. natural).

We also hypothesize that children will be more likely to indicate that natural agents are real, as compared to fictional and supernatural agents; and that such differences will increase with age [see 49, 50]. To test this, we will examine the main effect of agent type (supernatural vs. natural, fictional vs. natural) on responses to the reality question ("Is/are [agent] real?"), as well as statistical interactions between participant age (4–10 years) and agent type.

Drawing on past literature, we further predict that children's reality status judgments will be associated with their familiarity judgments [54]. More specifically, we expect that children will be less likely to attribute reality status to agents that are unfamiliar to them.

Additional exploratory analyses will examine relationships among reality judgments, familiarity judgments, and the emotional valence of the agent, and other higher-order statistical interactions. Beyond this, we plan to compare responses across samples from different cultural-religious groups; to compare children's responses to those of parents/caregivers, to explore relationships between children's responses and information provided in the Parent Survey; and to explore relationships between this and other tasks included in the Child Protocol (e.g., correlations between reality status judgments and questions about whether a given agent is good or bad).

**Property Attributions task: Assessing the role of folk physics, biology, psychology, and sociology in reasoning about religious agents.** The Property Attributions task is designed to evaluate conceptualizations of religious and supernatural agents [for similar tasks, see 8,38,55]. This is part of the Child Protocol; see Fig 2.

*Task design and procedure.* The Property Attributions task consists of a 20-item close-ended battery intended to assess participants' representations of the physical, biological, cognitive-epistemic, social-emotional, and sociological properties and constraints of two to three religious/supernatural agents, as compared to the properties and constraints of ordinary human agents.

The task features the "Biggest" god [51] in the cultural-religious setting as well as 1–2 additional prominent religious agents. These religious/supernatural agents are also featured in the Sorting task (described in the previous section) and in the Justifications task (described in a later section); if a child indicates that they are not familiar with one of these agents in the Sorting Task, this agent is omitted from the Property Attributions and Justifications tasks for that child. Questions about human or human-like prophets and other agents who are not the "Biggest" god are referred to as "Agent 1" for the purposes of this task, and always precede questions about the "Biggest" god ("Agent 2"). Throughout the task, these religious/supernatural agents are compared to an ordinary human ("a person"). See Table 2 for items selected for each US sample.

**Table 2. Sample-specific item selection for the Property Attributions and Justifications tasks, for the five religions included in US samples.**

| Agent | Religion | | | | |
|---|---|---|---|---|---|
| | *Protestantism* | *Catholicism* | *Church of Jesus Christ of L.D.S.* | *Judaism* | *Islam* |
| 1 | Jesus | Jesus | Jesus | Elijah | the Prophet Muhammad |
| 2 | God | God | God | God[a] | Allah |
| 3 | the Holy Spirit | the Holy Spirit | the Holy Spirit | Jesus | Jesus |

Unaffiliated children in the US are shown the version of the protocol that corresponds to the religion with which they are likely most familiar (see main text for details). See main text for descriptions of network-wide guidelines for item selection. "L.D.S." stands for "Latter-day Saints."

[a] If the researcher becomes aware that the child is more familiar with the term "Hashem" or "Adonai" then the more familiar term is used in place of "God" throughout the Child Protocol.

The 20 items used in this task probe five key domains of foundational laws that govern human behavior. For each domain, half of the four items are phrased as choices between two alternatives, and half are phrased as choices between an alternative and its negation; half are phrased in the generic, present tense and half in the hypothetical, subjunctive tense; and half present the violation of the law first and half present it second.

In the *physical* domain, four items probe reasoning about the constraints of physical objects: namely, the laws of solidity ("Would [agent] need to find a door in order to pass through a wall, or could [agent] pass straight through a wall without using a door?"), continuity ("Is [agent] ever in different places at the same time, or is [agent] always in one place at a time?"), mass ("Does [agent] have a shadow, or does [agent] have no shadow?"), and time ("Could [agent] go back in time and do last week over again, or could [agent] not do that?").

In the *biological* domain, four items probe reasoning about the constraints of biological animals: the need for food ("Could [agent] keep going without ever eating any food, or would a [agent] need to eat food eventually?"), susceptibility to illness ("Does [agent] ever get sick or does a [agent] never get sick?"), the need for sleep ("Would [agent] ever need to sleep, or would a [agent] never need to sleep?"), and the aging process ("Does [agent] stay the same age forever, or does a [agent] get older every year?").

In the *cognitive-epistemic* domain, four items probe reasoning about the constraints of agents with limited minds: fallibility/mistakes ("Does [agent] ever make mistakes or does a [agent] never make mistakes?"), limited memory ("Would [agent] remember everything forever or would a [agent] ever forget something?"), limited knowledge ("Does [agent] know everything or does a person not know everything?"), and limited ability to read others' minds ("Could someone trick [agent] by telling a lie, or would [agent] know that they were lying?").

In the *social-emotional* domain, four items probe reasoning about the constraints of beings who have social and emotional lives: susceptibility to loneliness ("Would [agent] be okay if a [agent] were all alone forever, or would a [agent] get lonely?"), susceptibility to anger ("Would [agent] ever get angry, or would [agent] never get angry?"), susceptibility to worry ("Is [agent] always sure that things will be ok, or does [agent] ever feel worried?"), and susceptibility to shyness ("Does [agent] ever feel shy, or does a [agent] never feel shy?").

Finally, in the *sociological* domain, four items probe reasoning about the constraints of beings who live in human-like social structures: kinship relations ("Does [agent] have a family, or does [agent] have no family?"), differential liking ("Does [agent] like everyone the same, or does [agent] like some people more than others?"), being part of a hierarchical power structure ("Would [agent] always be in charge no matter who else was around, or is there anyone who could tell [agent] what to do?"), and being subject to rules and norms ("Would [agent] ever have to follow any rules, or would [agent] never have to follow any rules?"

To introduce the task, the researcher tells the child, "Now I'm going to ask you some new kinds of questions. I'm going to ask you some questions about people, and I want you to think about normal, everyday people. I might also ask you some questions about others, like [Agent 1] and [Agent 2] (*if applicable*: and [Agent 3]). I might ask you about some things you said are not real, and I might ask you some questions that you think are kind of weird or silly, and that's okay. I'm still going to ask you, and I want you to take your best guess."

The 20 questions just described are then presented in random order. For each question, children are first asked to evaluate the human, and then the supernatural or religious agents (e.g., "Does a person stay the same age forever, or does a person get older every year? What about [Agent 1]?," "What about [Agent 2]?," and (if applicable), "What about [Agent 3]?"). If additional scaffolding is needed, in particular for younger children, the researcher repeats the question in full (e.g., "Does [Agent 1] stay the same age forever, or does [Agent 1] get older every year?").

In the US samples, this task is estimated to take 4 to 12 minutes with younger participants and 8 to 13 minutes with the oldest participants.

*Adaptation for US samples*. The final set of agents featured in the Property Attributions and Justifications tasks for the five religions included in US samples is presented in Table 2.

*Parallel task for parents/caregivers*. Parents/caregivers complete a slightly modified version of the Property Attributions task. Parents/caregivers answer the same 20 questions as children and provide certainty ratings for each question on an 11-point scale ranging from [0] "Not sure at all" to [10] "Extremely sure."

*Planned analyses*. We hypothesize that children will attribute more human-like properties to the human agent than to the supernatural/religious agents [see 8, 38, 55, 56]; i.e., that children will be more likely to endorse violations of the physical, biological, cognitive-epistemic, emotional, and sociological laws for the supernatural/religious agents than for the human. To test this hypothesis, we will examine the main effect of agent type (supernatural/religious vs. human).

We also expect that the difference between supernatural/religious and human agents will vary across domains; in particular, we hypothesize that the difference between the supernatural/religious agents and the human will be greater for violations of physical and biological laws than for violations of cognitive-epistemic, emotional, or sociological laws [56–58]. To test this hypothesis, we will examine statistical interactions between agent type (supernatural/religious vs. human) and domain (physical, biological, cognitive-epistemic, emotional, sociological).

Developmentally, we hypothesize that attributions of human-like properties to supernatural/religious agents will decrease with age [see 8, 49, 50, 55, 59, 60]. We anticipate that older children will be more likely to say that supernatural/religious agents will violate constraints, as compared to younger children; to test this hypothesis, we will examine the main effect of participant age (4–10 years), considering only supernatural/religious agents. We also anticipate that children's perceptions of the *difference* between the supernatural/religious and the human will increase with age; to test this, we will also examine a statistical interaction between agent type (supernatural/religious vs. human) and participant age (4–10 years).

Many of these analyses will be informed by assessments of the observed reliability (i.e., internal consistency) of items within each of the five domains (physical, biological, cognitive-epistemic, emotional, sociological).

Beyond these predictions, we plan to examine higher-order statistical interactions (e.g., an interaction between type, domain, and participant age); to characterize children's perceptions of the properties of specific agents (e.g., Jesus, God, etc.); to compare responses across samples from different cultural-religious groups; to compare children's responses to those of parents/caregivers; to explore relationships between children's responses in this task and information provided in the Parent Survey; and to examine relationships between this and other tasks included in the Child Protocol.

### Research question 2: How do children represent and reason about religion as an aspect of social identity?

**Religious Indicators task: Assessing knowledge of and self-identification with indicators of religious identity and affiliation.**  Building on previous work on children's understanding of the development of ethno-religious identity [e.g., 61–63], the Religious Indicators task is designed to assess children's familiarity with various indicators of religious group membership and their own sense of similarity to people who are associated with these indicators. This task is part of the Child Protocol; see Fig 2.

*Task design and procedure.* The Religious Indicators task consists of two blocks presented in a fixed order, with other tasks administered between the two blocks; see Fig 2.

In each block, children receive six test trials presented in a random order. On each trial, the researcher introduces a character and tells the child one property or characteristic about that character. On half of trials, the property paired with the character includes something that indicates that the character is a member of the target religious group. On the other half of trials, the property includes information that is intended to be neutral with respect to religious identity. (See Table 3 for properties selected for each US sample.) Each character is depicted with a grayscale, child-like silhouette. There is no visual depiction of the specific property provided, and silhouettes are matched with properties at random. The gender of the character is matched to the participating child's gender; if the participating child does not identify as a boy or a girl, or if the participating child's gender is unknown, the child is shown the "girl" version of the task.

As in other tasks, the range of religious and neutral properties included in this task is intended to vary across samples in order to preserve cultural and religious specificity and appropriateness, within certain limits intended to maintain a balanced level of standardization

**Table 3. Sample-specific item selection for the Religious Indicators task, for the five religions included in US samples.**

| | Religion | | | | |
|---|---|---|---|---|---|
| Property | *Protestantism* | *Catholicism* | *Church of Jesus Christ of L. D.S.* | *Judaism* | *Islam* |
| **Belief** | | | | | |
| *Religious* | Here's a kid who believes that Jesus died for our sins. | Here's a kid who believes that Jesus died for our sins. | Here's a kid who believes that Joseph Smith was God's prophet. | Here's a kid who believes that God sent ten plagues. | Here's a kid who believes that the Prophet Muhammad is Allah's messenger. |
| *Neutral* | Here's a kid who believes that George Washington was a US president. | Here's a kid who believes that George Washington was a US president. | Here's a kid who believes that George Washington was a US president. | Here's a kid who believes that George Washington was a US president. | Here's a kid who believes that George Washington was a US president. |
| **Holiday** | | | | | |
| *Religious* | Here's a kid who goes to church on Easter. | Here's a kid who goes to mass on Ash Wednesday. | Here's a kid who goes to church on Pioneer Day. | Here's a kid who visits the temple or synagogue during Hanukkah. | Here's a kid who celebrates Eid Al Fitr. |
| *Neutral* | Here's a kid who gives out valentines on Valentine's Day. | Here's a kid who gives out valentines on Valentine's Day. | Here's a kid who gives out valentines on Valentine's Day. | Here's a kid who gives out valentines on Valentine's Day. | Here's a kid who gives out valentines on Valentine's Day. |
| **Prayer-like practice** | | | | | |
| *Religious* | Here's a kid who says 'in Jesus' name, amen' when [she/he] is done praying. | Here's a kid who prays the rosary. | Here's a kid who says 'In the name of Jesus Christ, amen' when [she/he] is done praying. | Here's a kid who starts blessings by saying 'Baruch Atah Adonai' or 'Baruch Atah Hashem' [or if needed: 'Blessed are You O Lord']. | Here's a kid who does five daily prayers. |
| *Neutral* | Here's a kid who says good night when [she/he] goes to bed. | Here's a kid who says good night when [she/he] goes to bed. | Here's a kid who says good night when [she/he] goes to bed. | Here's a kid who says good night when [she/he] goes to bed. | Here's a kid who says good night when [she/he] goes to bed. |
| **Block 2** | | | | | |
| *Target and non-target religious group labels* | Christian or not Christian | Catholic or not Catholic | a member of the Church of Jesus Christ or not a member of the Church of Jesus Christ | Jewish or not Jewish | Muslim or not Muslim |

Unaffiliated children in the US are shown the version of the protocol that corresponds to the religion with which they are likely most familiar (see main text for details). See main text for descriptions of network-wide guidelines for item selection. "L.D.S." stands for "Latter-day Saints."

across samples. To achieve this, for each sample we include one religious property concerning an internal *belief state* that members of the target religious group are likely to hold, as well as a matched neutral belief that both members and non-members are equally likely to hold; one religious property concerning a *holiday* that members of the target religious group are likely to observe or celebrate, as well as a matched neutral holiday that both members and non-members are equally likely to celebrate; and (3) one religious property concerning a *prayer-like practice* (e.g., prayer, blessings, or other communication with the divine) that members of the target religious group are likely to engage in, as well as a matched neutral activity that both members and non-members are equally likely to engage in habitually.

The properties included in this task sometimes focus on the character individually (e.g., "Here is a kid who. . .") and sometimes on the character's family (e.g., "Here is a kid whose family. . ."), as deemed appropriate in a given setting. Research teams have been encouraged, but not required, to select religious properties that differentiate among salient religious groups in the local context (e.g., choosing a belief about Jesus rather than God in settings where Christians live alongside members of other monotheistic religions); the only strict requirement is that these properties should be salient aspects of religious identity and should be familiar and relevant to children, as confirmed by local informants (e.g., religious leaders, parents of young children).

The six properties described above (three religious, three neutral) are used in both blocks of the task. The first block focuses on children's sense of similarity to the character featured on each trial. The researcher introduces the block by saying, "I'm going to tell you about some kids, and I want to know if they're like you or different from you. There are no right or wrong answers here—I just want to know what you think." On each trial, the researcher shows the child the silhouette of the character and reads the corresponding property aloud to the child. The child is then asked two questions in a fixed order: "Is [she/he] like you or different from you?" and "A little bit [like you/different from you] or a lot [like you/different from you]?" These two questions are designed to be combined into a 4-point response scale ("a lot different from me," "a little bit different from me," "a little bit like me," "a lot like me").

The second block follows an identical design but focuses on the child's assessment of whether the characters are or are not members of the target religious group. The researcher introduces the block by saying, "Now I'm going to show you those kids again, and this time I want to know something different. I want to know whether they are [target religious group label] or *not* [target religious group label]. It's ok if you're not sure—I just want you to make your best guess." As in the first block, on each trial, the researcher shows the child the silhouette of the character and reads the corresponding property aloud to the child. The child is then asked two questions in a fixed order: "Is [she/he] [target religious group label] or *not* [target religious group label]?" and "Probably [target religious group label/*not* target religious group label] or definitely [target religious group label/*not* target religious group label]?" Again, these two questions are designed to be combined into a 4-point response scale.

Depending on the sampling plans in place at a given field site, the target religious group featured in this task sometimes corresponds to the participating child's own religious group membership, sometimes corresponds to a religious group with which the child's parent has indicated that they are familiar, and sometimes corresponds to a religious group that is prevalent in the participating child's local context. The target religious group featured in this task is matched to the religious "target group" or "reference group" featured in the Sorting, Child as Transmission Agent, Social Essentialism, and Norm Violations tasks (described in other sections).

The question about the religious identity of the character is phrased according to local norms about how to refer to the religious group in question, both in terms of syntax (e.g., "Jewish" vs. "a Jew") and in terms of specificity (e.g., "Christian" vs. "Protestant"/"Catholic").

In the US samples, each block is estimated to take 2 to 4 minutes across the age range, for a total of 4 to 8 minutes for the task as a whole.

*Adaptation for US samples*. The final set of properties featured in the Religious Indicators task for the five religions included in US samples is presented in Table 3.

*Parallel tasks with parents/caregivers*. There is no parallel task for parents/caregivers.

*Planned analyses*. Our first set of hypotheses concerns the second block of the task, in which children are asked to infer the religious group membership of characters ("Is [she/he] [target religious group label] or *not* [target religious group label]?"). First, we hypothesize that children will judge characters who are associated with the three religious properties included in this task as more likely to be members of the target religious group than characters who are associated with the three neutral properties [see 61, 63]. In other words, children will recognize the beliefs, holidays, and prayer-like practices included in this task as indicators of religious group membership. To test this hypothesis, we will examine the main effect of property type (religious vs. neutral) on children's responses in the second block of this task.

We also expect that children's responses to religious vs. neutral properties will vary across domains (e.g., that the difference between ascriptions of religious vs. neutral beliefs might function as a more powerful indicator of religious group membership than the difference between behaviors related to religious vs. neutral holidays) [see 64, 65]. To test this hypothesis, we will examine statistical interactions between property type (religious vs. neutral) and indicator type (belief, holiday, prayer-like practice) on children's responses in the second block of the task.

Developmentally, we hypothesize that children's tendency to recognize religious indicators and infer religious group membership will increase with age [65–67]. To test this hypothesis, we will examine a statistical interaction between property type (religious vs. neutral) and participant age (4–10 years) on children's responses in the second block of this task.

We will conduct a parallel set of analyses for data from the first block of this task, in which children are asked to judge how similar the characters are to themselves ("Is [she/he] like you or different from you?"). These analyses will assess whether children judge characters associated with religious vs. neutral properties to be more similar to themselves, and whether this tendency differs across specific religious indicator types (belief, holiday, prayer-like practice) or across the age range of our samples. These analyses will be exploratory; we do not have strong predictions about the strength or direction of these results.

Beyond this, we plan to examine higher-order statistical interactions (e.g., an interaction between property type, indicator type, and participant age); to conduct within-subjects analyses of the relationships between children's responses in the first and second blocks of this task (i.e., between children's similarity judgments and their recognition of religious indicators); to compare responses across samples from different cultural-religious groups; to explore relationships between children's responses in this task and information provided in the Parent Survey; and to examine relationships between this and other tasks included in the Child Protocol.

**Social Essentialism task: Assessing the "essentialization" of religious groups relative to other social groups (gender, wealth).** Drawing on previous research [e.g., 64, 68, 69], the Social Essentialism task is designed to assess the extent to which children essentialize social groups based on religion, gender, and wealth. This task is part of the Child Protocol; see Fig 2.

*Task design and procedure*. The Social Essentialism task consists of three blocks, presented in random order, focusing on three social categories: religion, gender, and wealth. Within each block, children are shown silhouettes of two groups of people, presented side by side, and identified verbally by the researcher as being members of contrasting social categories. Each block features one "reference group" (generally either the child's ingroup, a majority group in the local setting, or a group of higher social status in the local setting) and one "non-reference

group" (generally the child's outgroup, a minority group, or a group of lower social status). Throughout the block, the non-reference group is always mentioned first, and the reference group is always mentioned second; this applies to the introductions to each block as well as all test trials.

For the *religion* block, the reference and non-reference groups vary across samples. Depending on the sampling plans in place at a given field site, this sometimes corresponds to the participating child's own religious group membership, sometimes corresponds to a religious group with which the child's parent has indicated that they are familiar, and sometimes corresponds to a religious group that is prevalent in the participating child's local context. The reference religious group featured in this task is matched to the religious "target group" or "reference group" featured in the Sorting, Religious Indicators, Norm Violations, and Child as Transmission Agent tasks (described in other sections). The selection of the non-reference religious group depends on the selection of the reference religious group: In some samples, this yields a clear ingroup vs. outgroup contrast for participating children, while in other samples, the contrast is between a majority vs. minority religious group, or between a higher- vs. lower-status religious group.

For the *gender* block, the reference group is the participating child's own gender (e.g., "girls"), and the non-reference group is the contrasting binary gender ("boys"). If the participating child does not identify as a boy or a girl, or if the participating child's gender is unknown, the child is shown the "girl" version of the task.

For the *wealth* block, the reference group is always "rich people," and the non-reference group is always "poor people."

To introduce the task, the researcher says, "Let's talk about whether people are similar or different. Sometimes children tell me that two groups of people are a lot like each other, sometimes children tell me that two groups of people are really different from each other. I just want to know what you think. So I'll ask you a few questions about how people are the same or different on the inside and on the outside. You can say yes, or maybe, or no to any of my questions."

To introduce each block, the researcher says, "This time it's about [non-reference group] and [reference group]. All of these people over here [are from the same religion/are the same gender/have about the same amount of money]: They are all [non-reference group] [*researcher points to silhouettes on the left*]. And all of these people over here [are from the same religion/are the same gender/have about the same amount of money]: They are all [reference group] [*researcher points to silhouettes on the right*]."

Next the researcher asks four questions in random order: (1) "Are people born that way? Like, [non-reference group] are born [non-reference group] and [reference group] are born [reference group]?", (2) "Is it possible to tell whether a person is [non-reference group] or [reference group] just by looking inside their body, like by looking at their blood and bones?", (3) "Are [non-reference group] people's souls different from [reference group] people's souls?", and (4) "Is it possible for [non-reference group] to become [reference group]?" When the child has answered all four questions in that block, the researcher proceeds directly to the next block of the task, beginning with the introduction of the next two social groups.

In the US samples, this task is estimated to take 3 to 6 minutes across all age ranges included.

*Adaptation for US samples*. The final set of reference and non-reference religious groups featured in the Social Essentialism task for the five religions included in US samples is presented in Table 4.

*Parallel tasks with parents/caregivers*. Parents/caregivers complete the religion block of the Social Essentialism task (but no other blocks of this task), and they are asked to indicate their

**Table 4. Sample-specific item selection for the Social Essentialism task, for the five religions included in US samples.**

| Group label | Religion | | | | |
|---|---|---|---|---|---|
| | *Protestantism* | *Catholicism* | *Church of Jesus Christ of L.D.S.* | *Judaism* | *Islam* |
| *Reference group (Religion Block)* | | | | | |
| *plural noun form* | Christians | Catholics | members of the Church of Jesus Christ | Jews | Muslims |
| *adjective form* | Christian | Catholic | a member of the Church of Jesus Christ | Jewish | Muslim |
| *Non-reference group (Religion Block)* | | | | | |
| *plural noun form* | Muslims | Muslims | Muslims | Christians | Christians |
| *adjective form* | Muslim | Muslim | Muslim | Christian | Christian |

Unaffiliated children in the US are shown the version of the protocol that corresponds to the religion with which they are likely most familiar (see main text for details). See main text for descriptions of network-wide guidelines for item selection. "L.D.S." stands for "Latter-day Saints."

level of certainty for each response on an 11-point scale ranging from [0] "I am not sure at all" to [10] "I am extremely sure."

*Planned analyses*. Based on previous work [64, 68, 69], we hypothesize that we will observe the highest levels of social essentialism in the gender block and the lowest levels of social essentialism in the wealth block, and that social essentialism in the religion block will be intermediate between these extremes. To test this hypothesis, we will examine the main effect of domain (gender, religion, wealth) on children's tendency to select "essentialist" responses.

Developmentally, we hypothesize that these differences in social essentialism across domains will increase with age [see, e.g., 68]. To test this hypothesis, we will examine statistical interactions between domain (gender, religion, wealth) and participant age (4–10 years).

In testing these predictions, we plan both to conduct item-level analyses (e.g., examining differences across the four questions included within each domain, or examining domain differences considering each of these four questions separately); and to conduct analyses in which we combine responses across items.

Beyond these predictions, we plan to compare the responses of boy and girl participants, especially in the gender block; to compare responses of children from families varying in socioeconomic status, especially in the wealth block; to compare responses across samples from different cultural-religious groups in all blocks; to compare children's responses to those of parents/caregivers; to explore relationships between children's responses in this task and information provided in the Parent Survey; and to examine relationships between this and other tasks included in the Child Protocol.

**Norm Violations task: Assessing the perceived severity, alterability, and scope of religious norms relative to other norms (moral, conventional).** Building on previous work on children's understanding of religious norms [e.g., 70–72], the Norm Violations task is designed to assess children's familiarity with and reasoning about some of the most salient norms from their own religious group or a locally prevalent religious group, and to compare children's judgments of violations of religious norms to violations of moral and conventional norms. This task is part of the Child Protocol; see Fig 2.

*Task design and procedure*. To introduce the task the researcher says, "Sometimes people do things that are okay and sometimes people do things that are not okay. I'm going to tell you about things that some people did, and I want you to tell me if you think those things were okay or not okay. If it's okay, you can say 'okay,' or do this [*researcher demonstrates thumbs up, or another locally appropriate gesture*]. If it's not okay, you can say 'not okay,' or do this [*researcher demonstrates thumbs down, or another locally appropriate gesture*]."

The child is then presented with two practice trials: The first focuses on an action that is clearly not okay (insulting someone), and the other focuses on an action that is clearly okay (giving someone a present). These practice trials are intended to familiarize children with the format of the task; to encourage them to use the full range of response options available to them; to encourage them to take the task seriously (i.e., provide sincere responses even if the questions seem silly or unusual); and to establish that the task might include both stories where characters do things that are "okay" and stories where characters do things that are "not okay."

The main task consists of four blocks of questions presented in a random order.

At the beginning of each block, the child is presented with a grayscale silhouette of character. The researcher introduces the character as an adult member of the target religious group who habitually violates a religious, moral, or conventional norm; see Table 5 for norms selected for each US sample. There is no visual depiction of the specific norm violation, and silhouettes are matched with stories at random. The child is then asked a series of questions about the norm in question, in a fixed order. First, the child is asked about the *permissibility* of the norm violation ("Is that okay or not okay?") and is asked to explain their response ("Why is that [okay/not okay]?"). Second, the child is asked to rate the *severity* of the norm violation ("And how good/bad is it that. . .?") on a three-point scale ("not good/bad," "just a little bit good/bad," or "very good/bad"). Third, the child is asked about the *alterability* of the norm by religious authorities; this is gauged by asking the child to reassess the permissibility and severity of the norm violation if a religious authority declared the norm violation to be okay ("What if [God/another religious authority] said that it was okay to. . .–then would it be okay for. . .? And how good/bad would that be?"). Fourth, the child is asked about the *scope* of the norm, i.e., whether it applies beyond the target religious group; this is gauged by asking the child to reassess the permissibility and severity of the norm violation if it were committed by a non-group member ("This is a man named Morgan. Morgan is not Muslim. Morgan . . ."). The child answers all of these questions for a single norm before proceeding to the next block, which features a different norm.

After completing all four blocks of questions, the child is asked a series of questions intended to gauge awareness of religious norms in general and familiarity with the two religious norms featured in this task. First, the researcher asks the child, "Are there any rules that [target religious group members] should follow? Like anything they are supposed to do, or not supposed to do?" If the child does not provide a meaningful response, the researcher follows up with, "What are things that only people who go to the same [house of worship] as you do, or do not do?" Then, the researcher asks the child directly about each of the two religious norms featured in this task, by asking, "What about [target religious group members] [following the religious norm]?–is that something you have heard of?" If the child has already mentioned one or both of the religious norms in the initial open-ended question, the researcher skips the corresponding follow-up question(s).

Across the four blocks, we include two religious norms, one moral norm, and one conventional norm.

The two *religious* norms featured in this task conform to the following network-wide guidelines. First, all religious norm violations are transgressions of injunctive religious norms (i.e., violations of rules explicitly laid out by a religious authority) rather than transgressions of religious norms that might also be considered "moral" (e.g., being kind to others) or "conventional" (e.g., taking off shoes before entering someone's home). Second, all religious norms are intended to distinguish the target religious group from other world religions and other high-level religious groups in the local context; for example, for a version of the protocol designed for Muslim participants who also encounter Christians, both religious norms are required to

**Table 5. Sample-specific item selection for the Norm Violations task (including the familiarity assessment at the end of the task), for the five religions included in US samples.**

| | Religion | | | | |
|---|---|---|---|---|---|
| | *Protestantism* | *Catholicism* | *Church of Jesus Christ of L. D.S.* | *Judaism* | *Islam* |
| *Religious Norm Violation #1* | | | | | |
| Item | This is a [woman/man] named [character]. [Character] is Christian. [Character] never goes to church. [She/He] could, but [she/he] chooses not to. | This is a [woman/man] named [character]. [Character] is Catholic. [Character] never goes to church. [She/He] could, but [she/he] chooses not to. | This is a [woman/man] named [character]. [Character] is a member of the Church of Jesus Christ. [Character] never gives money to the church. | This is a [woman/man] named [character]. [Character] is Jewish. [Character] never lights candles on Friday nights. | This is a [woman/man] named [character]. [Character] is Muslim. [Character] never does [her/his] five daily prayers. |
| *Character, ingroup* | Christina/Mark | Christina/Mark | Christina/Mark | Rebecca/David | Yara/Danny |
| *Character, non-ingroup* | Jesse | Jesse | Jesse | Jesse | Jesse |
| *Religious Norm Violation #2* | | | | | |
| Item | This is a [woman/man] named [character]. [Character] is Christian. [Character] was never baptized—so a priest or pastor never put water on [her/his] head. | This is a [woman/man] named [character]. [Character] is Catholic. [Character] was never baptized—so a priest or pastor never put water on [her/his] head. | This is a [woman/man] named [character]. [Character] is a member of the Church of Jesus Christ. [Character] drinks coffee. | This is a [woman/man] named [character]. [Character] is Jewish. [Character] eats foods that are not kosher. | This is a [woman/man] named [character]. [Character] is Muslim. [Character] eats foods that are not halal. |
| *Character, ingroup* | Natalie/Luke | Natalie/Luke | Natalie/Luke | Rachel/Adam | Yasmina/Rami |
| *Character, non-ingroup* | Morgan | Morgan | Morgan | Morgan | Morgan |
| *Moral Norm Violation* | | | | | |
| Item | This is a [woman/man] named [character]. [Character] is Christian. [Character] hits other people for no reason. | This is a [woman/man] named [character]. [Character] is Catholic. [Character] hits other people for no reason. | This is a [woman/man] named [character]. [Character] is a member of the Church of Jesus Christ. [Character] hits other people for no reason. | This is a [woman/man] named [character]. [Character] is Jewish. [Character] hits other people for no reason. | This is a [woman/man] named [character]. [Character] is Muslim. [Character] hits other people for no reason. |
| *Character, ingroup* | Mary/Chris | Mary/Chris | Mary/Chris | Leah/Michael | Mona/Kareem |
| *Character, non-ingroup* | Jordan | Jordan | Jordan | Jordan | Jordan |
| *Conventional Norm Violation* | | | | | |
| Item | This is a [woman/man] named [character]. [Character] is Christian. [Character] wears socks on [his/her] hands. | This is a [woman/man] named [character]. [Character] is Catholic. [Character] wears socks on [his/her] hands. | This is a [woman/man] named [character]. [Character] is a member of the Church of Jesus Christ. [Character] wears socks on [his/her] hands. | This is a [woman/man] named [character]. [Character] is Jewish. [Character] wears socks on [his/her] hands. | This is a [woman/man] named [character]. [Character] is Muslim. [Character] wears socks on [his/her] hands. |
| *Character, ingroup* | Grace/Matthew | Grace/Matthew | Grace/Matthew | Sarah/Samuel | Leila/Nidal |
| *Character, non-ingroup* | Riley | Riley | Riley | Riley | Riley |
| *Religious authority* | | | | | |
| | God | God | God | God[a] | Allah |
| *Familiarity assessment* | | | | | |
| *Reference religious group* | Christians | Catholics | members of the Church of Jesus Christ | Jews | Muslims |
| *Norm #1* | going to church | going to church | giving money to church | eating food that is kosher | eating food that is halal |

*(Continued)*

**Table 5.** (Continued)

| | Religion | | | | |
|---|---|---|---|---|---|
| | *Protestantism* | *Catholicism* | *Church of Jesus Christ of L. D.S.* | *Judaism* | *Islam* |
| *Norm #2* | getting baptized | getting baptized | not drinking coffee | lighting candles on Friday night | praying five daily prayers |

Unaffiliated children in the US are shown the version of the protocol that corresponds to the religion with which they are likely most familiar (see main text for details). See main text for descriptions of network-wide guidelines for item selection. "L.D.S." stands for "Latter-day Saints."

[a] If the researcher becomes aware that the child is more familiar with the term "Hashem" or "Adonai" then the more familiar term is used in place of "God" throughout the Child Protocol.

pick out rules that Muslims abide by but Christians do not. (Religious norms are allowed but *not* required to distinguish sub-groups within an overarching religion, e.g., Sunni vs. Shia; Catholic vs. Protestant.) Third, religious norms are required to be salient and familiar to children, as confirmed by local informants (e.g., religious leaders, parents of young children); however, they are not required to be applicable to children themselves, i.e., they might focus on behaviors that are only required of adults.

Within these guidelines, the religious norms featured in this task are designed to vary across samples in order to preserve cultural and religious specificity and appropriateness. First, across samples, most religious norms apply equally to all genders, but some are gender-specific (e.g., applying only to women, or applying differently to men and women). If the norm is gender-specific, the gender of the character is fixed to be the relevant gender (regardless of the participating child's gender); otherwise, the gender of the character is matched to the participant's gender (or fixed to be a woman if the participating child does not identify as a boy or a girl, or if the participating child's gender is unknown). Second, across samples, some religious norms are prescriptive (i.e., outlining a behavior that a person should do), while others are proscriptive (outlining a behavior that a person should not do). Third, across samples, religious norms cover a wide range of religious content, including but not limited to rules surrounding eating, prayer, dress, and ritual.

In contrast to the variability present among the religious norms featured in this task, the moral and conventional norms are standardized across samples. The *moral* norm is "[Character] hits other people for no reason": an obvious and familiar violation of the common moral injunction against unjustified harm to others. The *conventional* norm violation is "[Character] wears socks on [his/her] hands" (or some close variant): an obvious violation of common rules about how clothing is intended to be worn and is typically worn. Research teams agreed these norms were relevant in all planned samples at the time of the design of the study protocol.

Depending on the sampling plans in place at a given field site, the target religious group featured in this task corresponds either to the participating child's own religious group membership, to a religious group with which the child's parent has indicated that they are familiar, or to a religious group that is prevalent in the participating child's local context. The target religious group featured in this task is matched to the religious "target group" or "reference group" featured in the Sorting, Religious Indicators, Social Essentialism, and Child as Transmission Agent tasks (described in other sections). The names of characters are consistent with the religious group membership or non-membership described in the stories (e.g., the name "Jesús" would be acceptable for a character who is described as "Christian," but not for a character who is described as "Jewish" or as "not Christian").

With the exception of gender-specific religious norms (described above), the gender of the characters featured in this task are matched to the participating child's gender; if the participating child does not identify as a boy or a girl, or if the participating child's gender is unknown, the child is shown the "girl" version of the task.

In the US samples, this task is estimated to take 7 to 14 minutes with younger participants and 7 to 11 minutes with the oldest participants.

*Adaptation for US samples*. The final set of properties featured in the Norm Violations task for the five religions included in US samples is presented in Table 5.

*Parallel tasks with parents/caregivers*. All parents/caregivers are asked two open-ended questions designed to elicit salient religious norms for children in these settings. Parents/caregivers are first told, "Now we have some questions about religious rules. Most religions in the world have rules about being kind to others and staying pure. In addition, many religions have other rules that people are expected to follow as part of their faith. For example, in some religions, people are required to wear particular clothing (either all of the time or under certain circumstances), to eat or to avoid eating certain foods, to pray in particular ways or at particular times, or to change their behavior during certain holidays." Then they are asked to provide open-ended responses to the following two questions: (1) "Of these kinds of rules, what ones do you focus on most with your child?" and (2) "Why should your children follow these rules?"

In some samples, including all US samples, parents/caregivers are also asked to complete a task parallel to the Norm Violations just described. Following the open-ended questions described above, parents/caregivers proceed directly to the permissibility and severity assessments of the two religious norms that their child has assessed, with the gender of the characters matched to the participating child's gender. Parents/caregivers assess only the two religious norms and provide only permissibility and severity judgments about a norm violation committed by a member of the target religious group. After the permissibility and severity assessments, parents/caregivers are also asked about how they would feel and what they would do if their own child violated the norm in question. Parents/caregivers complete all questions about the first religious norm before proceeding to the next religious norm, and norms are presented in a fixed order for all parents/caregivers within a sample.

*Planned analyses*. The main contribution of the data on norms will be the cross-cultural examination of how children develop conceptions of religious norms. As part of this examination, we also seek to compare children's conceptions of religious norms to their conceptions of moral norms against the harming of others and conventional norms about how to dress.

First, with respect to children's distinctions between religious and moral norms, we hypothesize that children will (1a) deem moral violations less permissible and more severely negative than religious violations, (1b) provide religious justifications more often when explaining judgments about religious norm violations and refer to victim welfare more often when explaining judgments about moral violations, (1c) be more likely to judge that religious norms are alterable by religious authorities than to judge that moral norms are alterable by religious authorities, and (1d) be more likely to apply moral norms to characters who are not members of the target religion than to apply religious norms to members of the target religion. (For previous work that has informed these hypotheses, see [70, 72–78].)

Second, with respect to children's distinctions between religious and conventional norms, we predict that children will (2a) provide religious justifications more often when explaining judgments about religious norm violations and refer to secular authorities (e.g., teachers, parents) more often when explaining judgments about conventional violations, and (2b) be more likely to apply conventional norms to characters who are not members of the target religion than to apply religious norms to members of the target religion. (See [70, 72–78].)

To test these hypotheses, we will examine the main effects of norm type (religious vs. moral, religious vs. conventional) on children's initial permissibility and severity ratings (relevant to hypothesis 1a); their open-ended justifications of their permissibility ratings (relevant to hypotheses 1b and 2a); their permissibility and severity ratings in the "alterability" portion of the block (relevant to hypothesis 1c); and their final permissibility and severity ratings for characters who are not members of the target religion (relevant to hypotheses 1d and 2b). For predictions about alterability and scope (hypotheses 1c, 1d, and 2b), we will also examine the main effect of norm type on the *differences* between children's initial and later responses, i.e., compare children's permissibility and severity ratings for characters who are not members of the target religion to their own initial permissibility and severity ratings for members of the target religion.

Of central interest will be differences in children's responses across age and across samples from different cultural-religious groups; these analyses will be exploratory and will likely hinge on modifying the analyses just described to include statistical interactions with age and/or sample.

Beyond these predictions, we plan to conduct item-level analyses (e.g., examining relationships and comparisons across the permissibility, severity, alterability, and scope questions for each of the individual norms included in the study, or examining norm type differences considering each of these questions separately); to compare children's responses to those of parents/caregivers; to explore relationships between children's responses in this task and information provided in the Parent Survey; and to examine relationships between this and other tasks included in the Child Protocol.

**Direct questions about religion.** Building on previous work on the development of religious and ethnic identity [e.g., 61, 79–82], we include a set of questions designed to directly assess children's explicit understanding of religion and their own religious identity. This is the final task in the Child Protocol; see Fig 2.

*Task design and procedure.* The task begins with the researcher saying, "The last thing I want to do is ask you a few questions about religion. I might use some words that you don't know, and if that happens that's totally okay, just let me know and we'll skip that one. I might also ask you some questions that seem really silly or obvious to you, or I might even ask you the same thing a few different times—that's because I'm talking to really little kids too and I need to ask it a few different ways." The child is then presented with a set of questions in a fixed order, proceeding roughly from most open-ended and general to most close-ended and specific.

First, the researcher asks two open-ended questions about religion in general: (1) "When I say the word 'religion,' what do you think of? Can you explain what 'religion' means?"; and (2) "There are lots of different religions in the world. Can you tell me all the religions you've heard of?"

Then, the researcher asks an open-ended question about the child's family's religious identity. In some cases, this question is phrased as, "What about your family: Does your family belong to a certain religion?"; in other cases, the question is instead phrased as, "Does your family believe in a certain religion?" Researchers are allowed to ask both questions if they deem it appropriate, and these judgments are generally made in real time on a case-by-case basis. If the child indicates that their family does belong to or believe in a religion, the researcher follows up with the question, "Which religion?" If the child indicates that their family does not belong to or believe in a religion, the researcher follows up with the question, "Do you mean they don't belong to a religion at all, or do you mean you don't know which religion?" If at any point the child indicates that they are not sure how to answer the question, the researcher can choose to ask a pair of additional follow-up questions: "What does your family

believe in? Does your family ever go to church, or synagogue, or temple, or mosque or anything like that?"

Regardless of children's responses, the researcher then proceeds to ask two close-ended questions about the child's religious identity. Both questions use the same format: "Are you [five or more specific religious groups, presented in a random order], or something else, or none of these?" In the first question, the specific religious groups are all superordinate-level religious groups: Children in all samples hear options corresponding to five of the major world religions ("Buddhist," "Christian," "Hindu," "Jewish," and "Muslim"); depending on the groups present in the local setting, some children also hear additional superordinate-level religious groups (e.g., "Sikh," "Druze," "Taoist"). In the second question, the specific religious groups are all subordinate-level religious groups (e.g., denominations or sects) and are selected to include roughly 5–8 locally salient groups (e.g., "Shia," "Sunni," "Protestant," "Catholic"). Research teams working with more than one religious group within a single geographical location are encouraged to include the same response options for all religious groups. Finally, the child is asked, "Is there anything else you want me to know about you and your family?"

In the US samples, this task is estimated to take 2 to 4 minutes across the age range.

*Adaptation for US samples*. Children in all US samples are administered the same adaptation of this task. In the first close-ended question, the superordinate-level groups provided as response options are: "Buddhist," "Christian," "Hindu," "Jewish," and "Muslim," with no additional sample-specific options; these are presented in a random order, followed by three additional options presented in a fixed order: "Something else," "I don't know," and "None of these things." In the second close-ended question, the subordinate-level groups provided as response options are "Catholic," "Protestant," "a member of the Church of Jesus Christ," "Reform," "Orthodox," "Sunni," and "Shia"; these are also presented in a random order, followed by "Something else," "I don't know," and "None of these things," which are presented in a fixed order.

*Parallel tasks with parents/caregivers*. There is no directly parallel task for parents/caregivers, although parents/caregivers are asked to provide a range of demographic information about themselves and their families; see "Parent Survey," below.

*Planned analyses*. Analyses of the open-ended questions included in this task (e.g., "When I say the word 'religion,' what do you think of? Can you explain what 'religion' means?") will be qualitative and exploratory in nature and are not described in this manuscript.

For the two close-ended questions about religious group membership, we hypothesize that children who are themselves members of a religious group will be more likely to select their own religious group than other religious groups, both when presented with a range of "superordinate" religious groups and when presented with a range of "subordinate" religious groups [see, e.g., 80]. To test this hypothesis, for each question we will examine the distribution of correct vs. incorrect responses, relative to what we would expect by chance (given the variable number of options presented to children in a given sample).

Developmentally, we hypothesize that children's tendency to correctly self-identify will increase with age [see 61, 79, 81–83], both when presented with a range of "superordinate" religious groups and when presented with a range of "subordinate" religious groups. To test this hypothesis, for each question we will examine the main effect of participant age (4–10 years) on the likelihood of correct responses.

Beyond these predictions, we plan to conduct within-subjects comparisons of the two questions (about superordinate vs. subordinate group membership); to compare responses across samples from different cultural-religious groups; to explore relationships between children's responses in this task and information provided in the Parent Survey; and to examine relationships between this and other tasks included in the Child Protocol.

### Research question 3: How are religious and supernatural beliefs transmitted within and between generations?

Some of the data for this research question will come from comparing individual children to their parents/caregivers on the shared tasks described under Research Questions 1–2 (above). Beyond this, we include several tasks that tap into particular aspects of the various transmission processes that might shape children's developing religious cognition and identity.

**Justifications task: Assessing self-reported information sources for religious beliefs and behaviors.** Building on previous work on children's representations of supernatural phenomena, the Justifications task is designed to evaluate children's understanding of their own "source knowledge" regarding the 2–3 religious agents featured in the Property Attributions task ([for similar tasks, see, e.g., 8, 38, 52, 53, 55, 84, 85]). This task is part of the Child Protocol; see Fig 2.

*Task design and procedure.* The Justifications task directly follows the Property Attributions task (described in an earlier section) and features the same set of agents included in that task—i.e., "normal, everyday people" as well 2–3 religious/supernatural agents (which we refer to in this manuscript as "Agent 1," "Agent 2," and, if applicable, "Agent 3").

The task consists of a separate block of questions for each agent. Blocks are presented in a fixed order matching the order of presentation of agents in the Property Attributions task; as in that task, the first agent to be discussed is "people," followed by Agent 1, Agent 2, and (if applicable) Agent 3.

Each block begins with a very brief introduction—"You just answered a lot of questions about [agent]"—followed by three open-ended questions presented in a fixed order: (1) "What else do you know about [agent]? You can tell me stories or any other things you know about [agent]"; (2) "Can you tell me more about how you know all of this about [agent]?"; and (3) "If you wanted to know more about [agent], how would you find out?"

The next set of questions within each block focuses in particular on children's judgments about the reality status of the agent. First, the researcher asks the child to make a *reality judgment*: "Remember when we were talking about things that are real or not real? Can you remind me what you think about this: Is/are [agent] real or not real?" Then the researcher asks the child to assess their *certainty* about this reality judgment by asking, "How sure are you that [agent] is/are [child's answer: real/not real]? A little bit sure, or really sure?" This is followed by an assessment of the child's *source knowledge* regarding this reality judgment: "How do you know that [agent] is/are [child's answer: real/not real]?" The final question in this set of questions is about the child's sense of *community consensus* regarding this reality judgment: "Would most people in your [city/town/village] agree with you and say that [agent] is/are [child's answer: real/not real]?"

Finally, the child is asked two additional questions designed to shed light on their construals of the agent: (1) a judgment about the *valence* of the agent, "Is/are [agent] good or bad?"; and (2) a judgment about the *corporeality* of the agent, "Does/Do [agent] have a body/bodies?" In contrast to other questions in the Child Protocol, researchers are encouraged *not* to press the child to make forced-choice responses to these questions; if a child gives a response like "in between" or "kind of," the researcher instead asks, "Can you tell me more about that?"

In the US samples, this task is estimated to take 2 to 18 minutes with younger participants and 6 to 14 minutes with the oldest participants.

*Adaptation for US samples.* As described earlier, the final set of agents featured in the Property Attributions and Justifications tasks for the five religions included in US samples is presented in Table 2.

*Parallel tasks with parents/caregivers.* There is no parallel task for parents/caregivers.

*Planned analyses*. Before describing our analysis plans, we note that analyses of the many open-ended questions included in this task will require the development of coding schemes to note the presence and extent of various kinds of content (e.g., identifying certain content as "religious"; counting mentions of "God" or other religious agents; coding different types of explanations) as well as various manners of speaking (e.g., noting the use of generic sentences). In particular, we anticipate adapting the coding scheme developed by Harris et al. [49] and further expanded by Davoodi et al. [53] to characterize different types of explanations that children might give over the course of this task, including elaborations (e.g., of causal processes or properties of the agent), discussions of the source of their knowledge (e.g., references to learning about the agent in school, from a family member, from a religious text), and references to encounters with the agent (e.g., recounting first- or second-hand personal experiences with the religious agent).

First, we hypothesize that, for all open-ended questions, older children will offer richer and more detailed examples, descriptions, and explanations of what they know and how they know it [see 53, 54, 86]. To test this hypothesis, we will examine the main effects of participant age (4–10 years) using any relevant coding schemes we have developed to quantify and characterize children's responses. We also anticipate examining the main effects of agent (people, Agent 1, Agent 2, and, if applicable, Agent 3), as well as statistical interactions between agent and participant age.

In terms of children's perceptions of community consensus, we hypothesize that children will perceive stronger community consensus when asked about a normal, everyday person than when asked about the two or three supernatural/religious agents, and that this difference will increase with age [see 53, 87]. To test these hypotheses, we will examine the main effect of agent as well as statistical interactions between agent and participant age, considering the question about community consensus alone.

In terms of children's source knowledge, we hypothesize that children will be more likely to name specific sources when asked about the two or three supernatural/religious agents than when asked about people; that mentions of specific sources will be inversely related to perceptions of community consensus (i.e., children will be more likely to name specific sources when discussing agents whose reality status they perceive to be controversial within their community); and that both of these effects will be exaggerated among older children [see 53]. To test these hypotheses, we will examine the main effect of agent, statistical interactions between agent and participant age, the main effect of community consensus, and statistical interactions between community consensus and participant age, considering the question about source knowledge alone.

Beyond these predictions, we plan to conduct exploratory analyses of all of the closed-ended questions that were not detailed in the current planned analyses; to compare responses across samples from different cultural-religious groups; to explore relationships between children's responses in this task and information provided in the Parent Survey; and to examine relationships between this and other tasks included in the Child Protocol.

**Child as Transmission Agent task: Open-ended storytelling about what information (religious and otherwise) is critical for a newcomer to the family.** The Child as Transmission Agent task is a qualitative, exploratory task designed to capture the ways in which children are themselves vectors of information transmission, religious and otherwise. This task is part of the Child Protocol; see Fig 2.

*Task design and procedure*. In the Child as Transmission Agent task, children are asked to imagine a child who is new to their neighborhood and wants to learn how to do things the right way.

The task is divided into two blocks: The first is focused on how to behave during a religious holiday, and the second is focused on how to behave in a home setting. These blocks are presented at different points in the Child Protocol; see Fig 2.

In each block, the task is introduced with the following script: "Let's imagine [character] is a new kid in your neighborhood who is younger than you who doesn't know what happens at your house." The gender of the character is matched to the gender of the participating child; if the participating child does not identify as a boy or a girl, or if the participating child's gender is unknown, the child is shown the "girl" version of the task. The character is referred to with a common name not associated with a particular religion in the local setting.

In the *religious holiday* block, children are then told, "Now let's say it's a special day like [religious holiday]. [Character] needs your help to remember what you do when celebrating this special holiday."

Children are then invited to provide open-ended instructions and explanations about the things that the character should do to observe the religious holiday the right way. These open-ended instructions are structured around two key rituals, practices, rules, or norms associated with that religious holiday; see Table 6 for items selected for each US sample. The researcher prompts the child to explain why the character should behave this way by asking, "Why should [character] do that?"

Next, the researcher asks, "Is there anything else you would tell [character] so she/he knows everything she/he needs to know to do things the right way when celebrating [religious holiday]?" This question is designed to elicit spontaneous references to religious practices or beliefs when transmitting information to a naive peer.

Lastly, the researcher asks, "If [character] has more questions about what to do when celebrating [religious holiday], what should she/he do?" This question is designed to elicit the salience of various information sources relevant to religion.

The *home* block follows a parallel structure. After the standard introduction to the task, children are told: "Let's say [character] was going to come inside your house for the first time. [Character] needs your help to figure out how to do things the right way when she/he comes over." Children are then invited to provide open-ended instructions and explanations about the things that the character should or shouldn't do when visiting the participating child's home for the first time, structured around two moments in an ordinary day that might be associated with religious ritual (e.g., what to do or not do when entering the house, meeting an adult, or preparing for dinner). As in the *religious holiday* block, children are then invited to offer the character additional tips and to suggest sources of additional information.

At the end of each block, children are asked whether the character is a member of the target religious group or not. Depending on the sampling plans in place at a given field site, the target religious group featured in this task might correspond to the participating child's own religious group membership, to a religious group with which the child's parent has indicated that they are familiar, or to a religious group that is prevalent in the participating child's local context. The target religious group featured in this task is matched to the religious "target group" or "reference group" featured in the Sorting, Religious Indicators, Social Essentialism, and Norm Violations tasks (described in other sections).

In the US samples, each block is estimated to take 2 to 5 minutes across the age range, for a total of 4 to 10 minutes for the task as a whole.

*Adaptation for US samples*. The final set of properties featured in the Child as Transmission Agent task for the five religions included in US samples is presented in Table 6.

*Parallel tasks with parents/caregivers*. There is no parallel task for parents/caregivers.

*Planned analyses*. Before describing our analysis plans, we again note that analyses of the many open-ended questions included in this task will require the development of coding

**Table 6. Sample-specific item selection for the Child as Transmission Agent task, for the five religions included in US samples.**

| | Religion | | | | |
|---|---|---|---|---|---|
| | *Protestantism* | *Catholicism* | *Church of Jesus Christ of L. D.S.* | *Judaism* | *Islam* |
| ***Both blocks*** | | | | | |
| *Character* | Taylor | Taylor | Taylor | Sam | Maya/Sami |
| *Target religious group* | Christian | Catholic | a member of the Church of Jesus Christ | Jewish | Muslim |
| ***Religious Holiday Block*** | | | | | |
| *Holiday* | Easter | Easter | Easter | Passover | Ramadan |
| *Item #1* | Is there anything Taylor should do or not do in the weeks before Easter? | Is there anything Taylor should do or not do in the weeks before Easter? | Is there anything Taylor should do or not do in the weeks before Easter? | What should Sam do for Passover seder (the special dinner at the beginning of Passover)? | What should Maya/Sami do if [she/he]'s invited for iftar? |
| *Item #2* | What should Taylor do on Easter Sunday? | What should Taylor do on Easter Sunday? | What should Taylor do on Easter Sunday? | What should Sam eat during the week of Passover? | What should Maya/Sami do if s/he comes over on Eid? |
| ***Home Block*** | | | | | |
| *Item #1* | What should Taylor do when [she/he] sits down at the table for dinner before eating? | What should Taylor do when [she/he] sits down at the table for dinner before eating? | What should Taylor do when [she/he] sits down at the table for dinner before eating? | What should Sam do when [she/he] gathers with other people for a meal? | What should Maya/Sami do after dinner before bedtime? |
| *Item #2* | What should Taylor do if [she/he] comes over on a Sunday? | What should Taylor do if [she/he] comes over on a Sunday? | What should Taylor do if [she/he] comes over on a Sunday? | What should Sam do if [she/he] comes over on a Friday evening? | What should Maya/Sami do if [she/he] came over on a Friday? |

Unaffiliated children in the US are shown the version of the protocol that corresponds to the religion with which they are likely most familiar (see main text for details). See main text for descriptions of network-wide guidelines for item selection. "L.D.S." stands for "Latter-day Saints."

schemes to note the presence and extent of various kinds of content (e.g., identifying certain content as "religious"; counting mentions of "God" or other religious agents; coding different types of explanations for why the character should or should not behave in a certain ways, as well as various manners of speaking (e.g., noting the use of generic sentences). These coding schemes have not yet been developed.

We hypothesize that older children will be more likely to spontaneously mention religious rules, practices, rituals and other religious content than younger children, and will offer richer and more detailed examples, descriptions, and explanations of this religious content [see 71, 72]. To test these hypotheses, we will examine the main effects of participant age (4–10 years) using any relevant coding schemes we have developed to quantify and characterize religious content in children's responses.

We also anticipate comparing responses in the two blocks of this task via examinations of the main effects of setting type (religious holiday vs. home) as well as statistical interactions between setting type and participant age.

Beyond these predictions, we plan to conduct item-level analyses (e.g., examining differences across questions about the two specific "moments" in the home); to compare responses across samples from different cultural-religious groups; to explore relationships between children's responses in this task and information provided in the Parent Survey; and to examine relationships between this and other tasks included in the Child Protocol.

**Caregiver-Child Conversation task: Conversations about existentially arousing topics.** Building on previous work on the transmission of religious and other beliefs from

parents to children [e.g., 85, 88, 89], the Caregiver-Child Conversation task is designed to assess the nature and content of conversations between children and their caregivers surrounding such "existentially arousing" topics as the origins of the world, illness and death, natural disasters, and intergroup conflict [90–92]. Because this research was designed and is being conducted during the COVID-19 pandemic, we include one question about COVID-19 in this task ("How do we protect ourselves from COVID-19?"); we consider this a timely example of an existentially arousing topic related to illness and death that is likely to have been quite salient during participants' lives. Future researchers using this research protocol might consider retaining this question to the extent that it is still relevant to children's lives in a given setting, or replacing this question with another salient example of an existentially arousing topic tailored to the specific time and place in which they are conducting their research.

The Caregiver-Child Conversation task is considered to be its own study component, separate from the Child Protocol; see Fig 2.

*Task design and procedure.* Across samples, the range of individuals who qualify as "caregivers" for this task varies from being limited solely to parents and legal guardians, to including grandparents and older siblings.

The task begins with the following instructions, directed to the caregiver: "We want to know how families talk about the following questions: Where does the world come from? Why do natural disasters happen? Why do some groups of people not get along with other groups of people? How do we protect ourselves from COVID-19? How do we get better when we are really seriously sick? What happens when we die? In this part of the study, we are asking you and the child participating in this study to talk about these questions together."

The instructions continue by outlining an example of how the task will proceed: "For each question, you will see a picture that has two or three sections, with one section missing (marked with a '?'). For example: 'How do trees grow?'" This question is illustrated with the picture provided in Fig 3a. The instructions continue, "The pictures are there to help start the conversation about the question, but you do not have to use them. For example, you could talk about: What happens in the missing section? How would you tell the whole story? You can discuss each question for as long as you would like. If you get to a question that you would prefer not to talk about, please feel free to skip it."

Next, the caregiver and child are asked to discuss the first topic: "Where does the world come from?" This is illustrated with the picture presented in Fig 3b. This is always the first topic presented to participants. The remaining topics are presented in a random order: "How do we protect ourselves from COVID-19?" (Fig 3c); "Why do natural disasters happen?" (Fig 3d); "What happens when we die?" (Fig 3e); "How do we get better when we are really seriously sick?" (Fig 3f); and "Why do some groups of people not get along with other groups of people?" (Fig 3g). The pictures accompanying these conversation topics are designed to be relatively neutral with respect to race, ethnicity, cultural-religious background, and geographical location.

As stated in the instructions, the participating caregiver and child can opt to spend as much time as desired on each topic or can opt to skip the topic altogether. The topics and their corresponding illustrations are standardized across samples.

In the US samples, this task is estimated to take 5 to 10 minutes across the age range.

*Adaptation for US samples.* In the US, only parents and legal guardians are eligible to participate as "caregivers" in this task. There are no restrictions placed on the number of caregivers permitted to participate together with a participating child, but most children participate in this task with a single parent/caregiver.

Caregivers in the US are given a few additional sample-specific instructions intended to clarify how we expect them to complete the task: "In this activity, we really want to get a sense

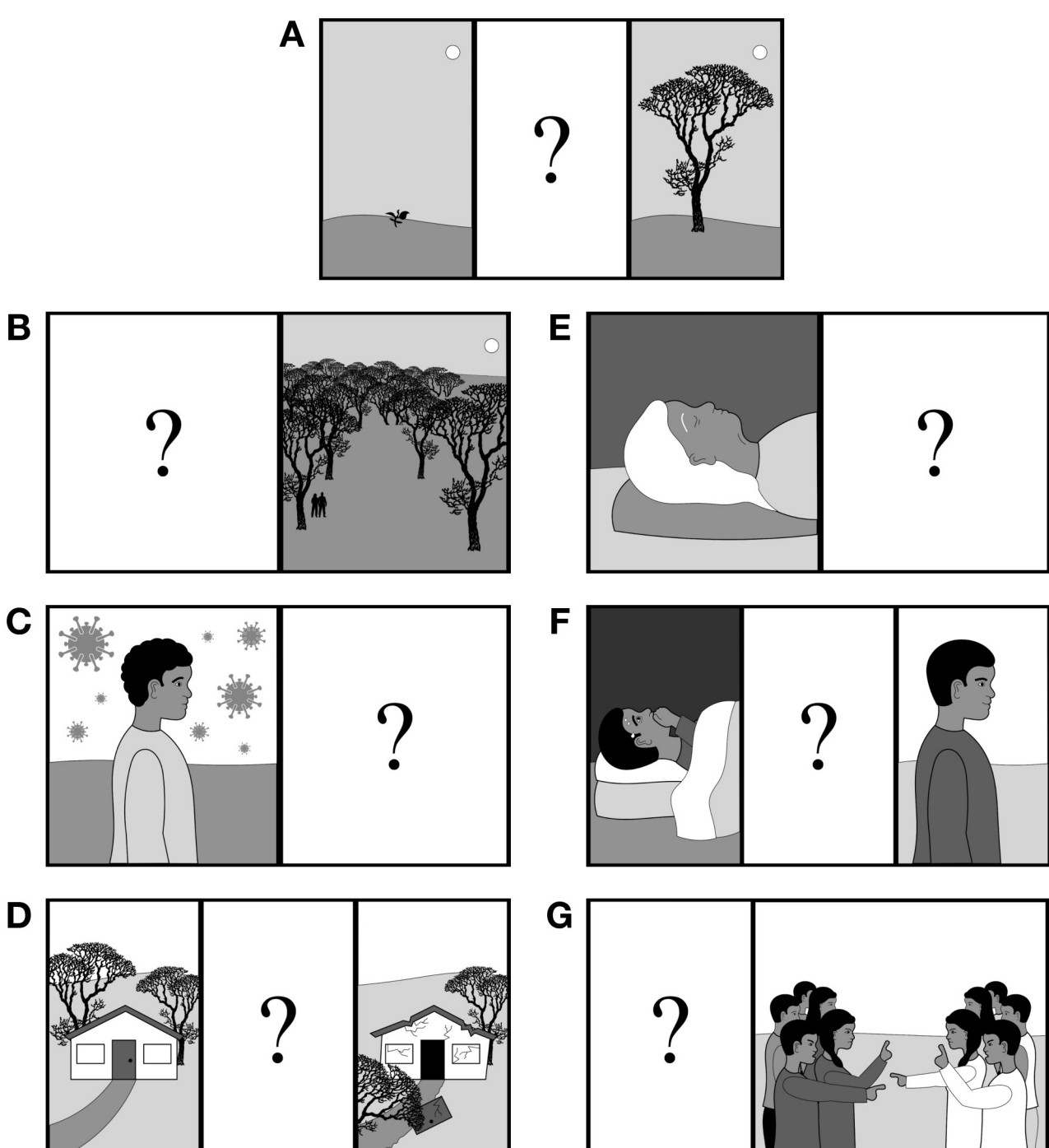

**Fig 3. Pictures used to illustrate the "existentially arousing" conversation topics included in the Caregiver-Child Conversation task.** (a) The example topic provided in the task instructions: How do trees grow? (b) Where does the world come from? (c) How do we protect ourselves from COVID-19? (d) Why do natural disasters happen? (e) What happens when we die? (f) How do we get better when we are really seriously sick? (g) Why do some groups of people not get along with other groups of people?".

of how you and your child talk about things together. We know in other activities we've asked you not to influence your child's responses to our questions. We are not worried about that here. In this activity, please just interact with your child the way you normally would. Next you'll see the list of all the questions that will come up in this activity. Don't start your conversations just yet. We just want you to know what kinds of questions may come up." Caregivers then proceed to the standard instructions and stimuli described in the previous section.

When we first began data collection in the US, caregivers were emailed instructions about how to complete and record this task independently and asynchronously without researcher supervision. At this point in data collection, we instead administer this task to caregivers and children via live video chat at the end of the Child Protocol. Analyses of data from this task will take into account differences across participants in the mode of study administration; in the US, this will include noting which caregivers completed the task independently vs. via live video chat with a researcher.

*Related tasks for parents/caregivers*. In the Parent Survey, parents/caregivers are asked to reflect on the Caregiver-Child Conversation task. For each of the six conversation topics included in this task, the survey-taker (i.e., the child's parent/caregiver) is asked how "natural" it would be for the child to discuss that topic with the caregiver who participated in the task; whether the participating caregiver chose to discuss or to skip that topic in the course of completing the task, and why/why not; whether the survey-taker had discussed that topic with that child before the child completed the task; and whether there was anything that surprised them about the child's conversation about that topic.

*Planned analyses*. Before describing our analysis plans, we note that analyses of these conversations will require the development of coding schemes to note the presence and extent of various kinds of content (e.g., identifying certain content as "religious"; counting mentions of "God" or other religious agents; coding different types of explanations) as well as various manners of speaking (e.g., noting the use of generic sentences, noting the presence or absence of question-asking, characterizing turn-taking between caregivers and children, characterizing conversations as predominantly caregiver-led or child-led). These coding schemes have not yet been developed.

We hypothesize that caregiver-child dyads including older children will engage in richer and more detailed conversations, and that these conversations will incorporate more (and more detailed) discussions of religious content [93–95]. To test these hypotheses, we will examine the main effects of participant age (4–10 years) using relevant coding schemes we develop to quantify and characterize the richness of these conversations and the presence of religious content.

We also anticipate comparing conversations across the six conversation topics included in this task via examinations of the main effects of topic, as well as statistical interactions between topic and participant age.

Beyond these predictions, we plan to compare responses across samples from different cultural-religious groups; to explore relationships between the content and nature of conversations elicited by this task and information provided in the Parent Survey; and to examine relationships between this and other tasks included in the Child Protocol.

**Parent survey.**   One of the primary means by which this study protocol addresses our third research question—How are religious and supernatural beliefs transmitted within and between generations?—is through an exhaustive survey administered to a parent or another primary caregiver of each child participating in the study.

As briefly described under "Parent Survey (overview)," above, this Parent Survey consists of a number of elements. As indicated in Fig 2, some portions of the Parent Survey are optional: Research teams may opt into including these elements in their adaptation(s) of the Parent

Survey depending on the interests of the team and on the anticipated feasibility of lengthening the survey for participating parents in a given sample.

Here we provide general descriptions of the Parent Survey; for the full text of the Parent Survey (as adapted for the five target religions included in US samples), please visit the following OSF repository: https://osf.io/dumf4/. The Parent Survey is its own study component, separate from the Child Protocol and Caregiver-Child Conversation task; see Fig 2.

*Survey design and procedure.* The Parent Survey begins with a variety of demographic questions about the survey-taker (i.e., the parent/caregiver), including age, country of birth, current state/province of residence, gender, race/ethnicity, number of years living in the country, cultural identity and influences, relationship status, highest level of formal education, and employment status. In some field sites and samples, including in all US samples, parents/caregivers also answer questions about their personal political affiliation/ideology.

The next set of questions focuses on parents/caregivers' own religious affiliation and religious experiences, including both open-ended and forced-choice responses to questions about their religious identity/affiliation. In some samples, including in all US samples, parents/caregivers who indicate that they are not currently religious also answer one question about atheism. Regardless of their responses to these initial questions, all parents/caregivers are then asked several questions intended to gauge how frequently they participate in various religious and spiritual practices, with examples tailored to the cultural-religious setting.

The next set of questions focuses on the language(s) that parents/caregivers use in their own daily and religious lives, how well they understand these languages, and the circumstances in which they use these languages.

The survey then shifts to questions about another important caregiver in the child's life. Survey-takers are first asked, "Other than you, who would you say plays the most important role in raising your child?" Then they are asked to provide information about this other caregiver, including age, gender, race/ethnicity, highest level of formal education, employment status, and religious affiliation.

Parents/caregivers are then asked a set of questions about their household, including what type of area they live in; how many people are in the household; the number, age, and gender of all of the survey-taker's own children; and the people that the participating child lives with (e.g., parents, siblings, grandparents, aunts, uncles, cousins).

Parents/caregivers then proceed to a set of questions intended to gauge the family's financial security and exposure to hardship. This set of questions was designed in collaboration with the research team led by Amanda Tarullo, Peter Rockers, and Denise Evans, who have extensive experience assessing financial security and exposure to hardship across a range of international field sites. Questions assess the parent/caregiver's subjective socioeconomic status via the MacArthur Scale of Subjective Social Status–Adult Version [96]; the potential impact of lost income; access to resources (e.g., running water, electricity, vehicles, technology, internet); experiences of financial insecurity, food insecurity, and access to medical care; and whether anyone close to the participating child has died during the child's lifetime.

The survey then shifts to questions about the participating child. Survey-takers are first asked to describe their own relationship to the child, and then asked to provide information about whether the child was born preterm; the child's age, gender, and race/ethnicity; and the child's educational experiences, including several questions intended to gauge the child's exposure to religious education (with examples tailored to the cultural-religious setting).

The next set of questions focuses on the participating child's exposure to religion. The parent/caregiver is asked several questions intended to gauge how frequently the child participates in various religious and spiritual practices, with examples tailored to the cultural-religious setting.

The next set of questions focuses on the participating child's exposure to language, including the language(s) that the child uses, hears, speaks, or reads in their daily and religious lives; when they started to learn these languages; how well they understand these languages; and the circumstances in which they use these languages.

The next part of the survey features two pre-existing measures of individual and cultural differences in beliefs about raising children, which (we speculate) might influence how parents/caregivers transmit religious beliefs and behaviors to their children. The first measure that parents/caregivers are asked to complete is the widely-used Authoritarianism scale [97, 98], in which parents/caregivers make forced-choice decisions about the most important qualities for a child to have; in our adaptation of this measure, the response options include the option, "I prefer not to choose one over the other" so that the scale is culturally appropriate for the wide range of sites included in this study. The second measure is the Family Cohesion scale [99], in which parents/caregivers indicate their agreement with a range of statements about their perceptions of support and togetherness within their family; in our adaptation of this measure, parents/caregivers provide agreement ratings on an 11-point scale ranging from [0] "I do not agree with this at all" to [10] "I completely agree with this." For both of these measures, items are presented in a random order for each participant.

The next part of the survey features two blocks of a newly developed measure of parental ethnotheories, cultural belief systems about how and from which sources children should learn. This measure is based on previous work, which has articulated a variety of dimensions of parental ethnotheories, such as how best to set up the environment for a child to focus on a particular topic of interest, which people in a child's life should provide this stimulation, and parents' perception of developmental timetables [see, e.g., 100–105]. The design of these questions was led primarily by Kirsten Lesage and Rebekah Richert. The two blocks of questions about parental ethnotheories are presented in a fixed order: The first focuses on the topic "religion," and the second focuses on the topic "social issues." For the social issues block, the phrase "social issues" is defined with the following examples: "Some examples might be poverty, homelessness, discrimination, and immigration." In each block, parents/caregivers are asked to rate their agreement with 17 statements presented in a random order (e.g., "It is important to me that my child thinks about [topic]"; "My child should ask questions about [topic]"; "My child should only learn about [topic] in our home," accompanied by the follow-up question, "Where else would it be okay for your child to learn about [topic]?"). Parents/caregivers provide agreement ratings on an 11-point scale ranging from [0] "I do not agree with this at all" to [10] "I completely agree with this." Following these 17 agreement ratings, parents/caregivers are asked four more questions about the topic: "At what age should children in general know about religion?"; "Whether or not it is ok, at what age do children typically start asking questions about [topic]?"; "What were the specific [aspects of topic] you had in mind when answering these questions?"; and "Is there anything else about your views on [topic] that you think would be important for us to know?"

The next part of the survey features a set of tasks designed to correspond to tasks included in the Child Protocol, in the following fixed order: Sorting task, Property Attributions task, Norm Violations task (including both open-ended questions and a subset of the trials included in the child version of the task), Social Essentialism task. The parent/caregiver is then asked to reflect on the Caregiver-Child Conversation task. For full descriptions of the parental versions of these tasks, see the corresponding task descriptions in earlier sections.

The next part of the survey features up to three pre-existing measures of individual and cultural differences in beliefs about raising children and more general orientations to thinking and reasoning, which (we speculate) might influence how parents/caregivers transmit religious beliefs and behaviors to their children. Research teams are free to include one, two, or all three

of these measures, or to opt out of including any of these measures. These three measures are presented in a fixed order (as relevant). The first measure is the Familism scale [106], in which parents/caregivers indicate their agreement with a range of statements about values related to family support, family interconnectedness, familial honor, and subjugation of self for family. The second measure is the Parental Support Questionnaire [107, 108], in which parents/caregivers indicate whether a series of statements were "usually true" or "usually false"; these statements are intended to assess the degree and kind of support parents/caregivers tend to provide to the participating child. The third measure is the Comprehensive Thinking Styles Questionnaire [109], in which parents/caregivers indicate their agreement with a range of statements about their own personal style of thinking. In our adaptation of the Familism scale and the Comprehensive Thinking Styles Questionnaire, parents/caregivers provide all agreement ratings on an 11-point scale ranging from [0] "I do not agree with this at all" to [10] "I completely agree with this." For all three measures, items are presented in a random order for each participant.

The next part of the survey features a newly developed set of questions about religious diversity and conflict in the child's local community. This set of questions was designed in collaboration with the research team led by Jocelyn Dautel, Laura Taylor, Aidan Feeney, John Coley, and Hannah Kramer, who have extensive experience assessing intergroup contact and conflict across a range of international field sites. This set of questions is only administered to parents/caregivers in a subset of samples (including all US samples). Parents/caregivers are first asked questions about their own religiosity relative to other adults in their local area, the prevalence of religiosity among people in their local area, and the diversity of religions present in their local area. They are then asked to identify up to six religious groups other than their own who are present in their local area. Participants then respond to a series of questions that tap into their perceptions of conflict between their own group and the other group(s), their contact (quantity and quality) with the other group(s), and their familiarity with the other group(s). Parents/caregivers provide all agreement ratings on an 11-point scale, with the labels provided for the endpoints [0] and [10] varying across items. Items are presented in a fixed order.

The final part of the survey features up to two pre-existing measures of individual differences in religiosity. Parents/caregivers in different sites receive different subsets of these measures, ranging from neither to both of them. In samples where one or both of these measures are included, only participants who have indicated in the initial question about religious affiliation that they themselves are currently religious or used to be religious are shown these measures. The first measure is the Intrinsic/Extrinsic Religious Orientation Scale [110, see also 111, 112], in which parents/caregivers indicate their agreement with a range of statements about why they engage in religious practices. The second measure is an adaptation of the Multi-Group Ethnic Identity Measure-Revised [113], which we have modified to focus on religious identity rather than ethno-cultural identity. In this scale, parents/caregivers indicate their agreement with a range of statements about their sense of identity, affiliation, and belonging to their religion. In our adaptation of these two measures, parents/caregivers provide all agreement ratings on an 11-point scale ranging from [0] "I do not agree with this at all" to [10] "I completely agree with this," and items are presented in a random order for each participant.

Parents/caregivers are required to respond to all questions in the Parent Survey but are always given the option to decline to respond to a particular question (e.g., a multiple-choice option stating, "I prefer not to answer"). Parents/caregivers are periodically given opportunities to expand on or clarify their responses via open-ended responses interspersed throughout the survey, as well as a general open-ended question at the end of the survey.

For all tasks parallel to tasks in the Child Protocol, the parent/caregiver sees the same version of the task that their child has seen.

*Adaptation for parents/caregivers of siblings.* In many samples, including all US samples, multiple siblings from the same family participate in the study. In this case, any parents/caregivers who have more than one child participating in the study completes the full Parent Survey for the first child to participate, and an abbreviated Parent Survey for any subsequent children. This "sibling survey" consists of an abbreviated set of demographic questions about the child's caregivers and household, the full set demographic questions about the child, the full set of questions about the child's exposure to religion and language, the full set of questions about parental ethnotheories, questions related to the Norm Violations and Caregiver-Child Conversation tasks, and the Parental Support scale (if included in the standard Parent Survey for that sample).

*Adaptation for US samples.* In all US samples, parents/caregivers are asked to complete all of the required and all of the optional elements in the Parent Survey. For the full text the Parent Survey (as adapted for the five target religions included in US samples), including sample-specific stimuli and examples, please visit the following OSF repository: https://osf.io/dumf4/.

*Planned analyses.* Because our primary purpose in this paper is to lay out the details and planned analyses of the Child Protocol, we do not detail planned analyses of the Parent Survey here.

## Additional tasks for children

**Memory Span tasks.** In addition to the tasks described above, which focus on the diversity and development of religious cognition in particular, the Child Protocol also includes a pair of tasks designed to tap into one aspect of general cognitive development: a pair of Memory Span tasks, which includes both *digit span* and *word span* versions.

Memory Span tasks are a widely used measure of executive function and cognitive development. A "forward" set of test trials, in which the child is asked to repeat a sequence of items in the same order that they hear it, is widely understood to assess the child's short-term auditory memory. A "backward" set of test trials, in which the child is asked to repeat a sequence of items in reverse order, is widely understood to assess the child's ability to perform mental manipulations of verbal/auditory information in short-term memory. For the purposes of this Child Protocol, we adapted the "Memory for Digit Span assessment: Digits Forward and Digits Backward" components of the Wechsler Intelligence Scales for Children-Revised (WISC-R) [114].

*Task design and procedure.* The Child Protocol includes two versions of the Memory Span task: one version featuring the numbers 1–9 (*digit span*), and one featuring a set of familiar words (*word span*). These tasks are presented in a random order, with other tasks administered between them (see Fig 2). Each version of the Memory Span task consists of two blocks: a *forward* block, in which the child is asked to repeat a set of items in the same order that they hear them, directly followed by a *backward* block, in which the child is asked to repeat the items in reverse order.

At the beginning of the first Memory Span task presented to a child (either *digit span* or *word span*), the researcher administers a newly designed introduction and set of practice trials. This is not the standard introduction for this task; instead, this introduction was designed to accommodate the diverse samples included in the DBN, which include children with a wide range of exposure to formal schooling (from no exposure to many years), differing levels of experience using and discussing numbers, differing levels of exposure to and first-hand experience with animals (the semantic category from which the words used in the *word span* version of the task have been drawn in all samples to date), and so on.

The researcher begins by introducing the task as follows: "I heard about a cool new game that these kids were playing in a town nearby and I want to play it with you. It's a game about how much you can remember." The framing of this task as a game was intended to provide some explanation for why children are asked to complete an otherwise meaningless task; network members consider this particularly important in settings where children have less experience being "tested" by adults. The researcher then explains the rules of the game for the *forward* block: "So here are the rules. I am going to do something. Your job is to do it in the same order that I did it."

In the next part of the introduction, the researcher guide the child through practice trials of the forward block, beginning with two practice trials involving parts of the physical body: "So, if I touch my nose [*researcher touches nose*] and then my forehead [*researcher touches forehead*], your job is to touch your nose [*researcher touches nose*] and then your forehead [*researcher touches forehead*]. Can you do that now just like I did?" The researcher then encourages the child to touch their own nose and forehead, corrects the child if needed, and then gives the child one more opportunity to practice in this physical modality. The intention behind grounding the child's understanding of the game in the physical, bodily modality is to give children in all samples a similar foundation for understanding the rules of the game rather than jumping immediately to verbal responses, which network members anticipate will be more familiar to children in settings with high exposure to formal schooling.

At this point in the introduction, the researcher asks the child to repeat the rules of the game: "Can you tell me what the rules of the game are?" Regardless of what the child says, the researcher responds, "The rule is: listen to me, then do what I do in the same order that I did it."

In the last part of the introduction, the researcher introduces the child to the specific version of the task at hand (*digit span* or *word span*), by saying, "So I heard one way those kids play this game was by saying [numbers/animals]. So I am going to say some [numbers/animals]. Your job is to say them back to me in the same order that I said them to you." Then the researcher administers 2–3 practice trials, in which the researcher affirms correct responses and corrects the child if needed.

The researcher then proceeds to the test trials for the forward block of test trials. These test trials follow the instructions for the "Memory for Digit Span assessment: Digits Forward" component of the WISC-R [114]. Within this block, there are up to eight "levels" of test trials, fixed in ascending order of difficulty from trials featuring sequences of two items each (Level 1) through trials featuring sequences of nine items each (Level 8). Each level includes two test trials, i.e., two sequences of items to be remembered; an additional two back-up trials are also available for each level in the case that the researcher or child is interrupted during a test trial (e.g., if the internet cuts out in an online test session). On each trial, the researcher reads a sequence of items aloud to the child at a steady rate of approximately one item every second. The researcher then pauses to allow the child to repeat back the sequence of items. The researcher does not offer any positive or negative feedback about the child's response, and incorrect initial responses are counted as incorrect even if the child subsequently corrects their own mistake. If the child gives a correct response for at least one of the two trials in the current level, the researcher proceeds to the next level. As soon as the child gives two incorrect responses in the same level, the forward block of test trials stops.

After the child completes the forward block of the first Memory Span game, the researcher gives an extended introduction to the backward block. Again, this custom introduction is designed to accommodate the diverse samples included in this study. The researcher begins by explicitly noting a change in the rules: "Okay. So that was fun! In the next part of the game, the rules change a little. This time, when I do something, your job is to do it backwards. So the last

thing I do, is the first thing you do. Let's practice." The rest of the introduction and the test trials are directly parallel to the introduction of the forward block described above.

For each child, the second Memory Span task (*word span* or *digit span*) features briefer versions of the introductions described above but is otherwise identical. For the forward block, the researcher introduces the rules as follows: "Remember that cool new game we were playing before about remembering [numbers/animals]? I'm going to remind you of the rules and then we can play it again. This time it's with [animals/numbers]. So I am going to say some [animals/numbers]. Your job is to say them back to me in the same order that I said them to you. Can you tell me what the rules of the game are? Yeah, the rule is: listen to me, then say what I say in the same order that I said it." For the backward block, the researcher introduces the rules as follows: "Now we're going to play the second part of the game like we did before. Remember, now the rules change: This time, when I say something you say it backwards. So the last [animal/number] I say is the first [animal/number] you say. Can you tell me what the rules of the game are? Yeah, the rule is: listen to me, then say what I say backwards. So the last [animal/number] I say, is the first [animal/number] you say." Before beginning the block of test trials, the researcher then administers 2–3 practice trials in the modality of the specific version of the task at hand (i.e., featuring animals or numbers), as described above.

For the *digit span* version of the Memory Span task, the items used in all samples are the digits 1 through 9 (pronounced in the language of study administration for each sample).

For the *word span* version of the Memory Span task, items vary across samples. For each sample, the nine words used in the task are from the same semantic set (i.e., the same taxonomic category); this is to ensure that, from the child's perspective, the "hypothesis space" of words to be remembered is constrained in similar ways across samples. Research teams are encouraged but not required to use "animals" as the taxonomic category; in practice, all samples to date have done this. In addition to the requirement that the nine words are from the same taxonomic category, they are also required to be familiar to children as young as 4 years of age, and to be (roughly) 1–3 syllables in length. Research teams are advised that the ideal set of words would consist entirely of one-syllable words (or, if that is not possible, entirely of two-syllable words), but are also counseled that familiarity to children should be prioritized over syllable count.

In the US samples, this task is estimated to take 3 to 7 minutes with younger participants and 4 to 10 minutes with the oldest participants.

*Adaptation for US samples*. In the US, the set of words used in the word span version of this task is based on the most commonly produced animal words among 30-month-old US English speakers in the CDI according to the Wordbank database as of January 2022 [115]: bird, fish, horse, dog, bear, cat, cow, pig, bug. These 9 words are in the top 11 animal words; we have attempted to avoid including pairs of words that sound too much alike (e.g., "dog" and "duck"), words that rhyme ("dog" and "frog"), and words that children might easily confuse ("duck" and "bird").

*Parallel tasks with parents*. There is no parallel task for parents.

*Planned analyses*. By default, analyses of the Memory Span tasks will follow the standards for analyzing "Memory for Digit Span assessment: Digits Forward and Digits Backward" components of the WISC-R [114], including standard practices for assessing individual differences as well as age-related (developmental) differences.

More broadly, we anticipate researchers applying a wide range of analysis techniques to the data generated by this task, including within-subjects analyses of the relationships between children's responses in the *digit span* and *word span* versions of this task; examinations of relationships between this and other tasks included in the Child Protocol; examinations of relationships between children's responses in this task and information provided in the Parent

Survey; assessments of the degree to which these tasks should or should not be considered valid measures of working memory in a given cultural-religious setting; and careful comparisons of samples from different cultural-religious groups, with an eye toward understanding how children's responses in these tasks might shed light on any observed similarities and differences across samples for other tasks in the Child Protocol.

## Other information

**Qualitative analyses.** This protocol was designed to be a mixed-methods study, yielding data that lends itself to a wide range of quantitative and qualitative analysis approaches. The "planned analyses" sections in the current manuscript have focused on some key quantitative analyses (albeit a subset of the many quantitative approaches that we anticipate network members and others applying to these datasets).

Here, we highlight that we consider qualitative analyses to be critical to understanding the multifaceted data generated by this research protocol. Such analyses will include, at a minimum, analyses of children's responses to the open-ended questions included in many of the tasks in the Child Protocol (in particular, the Justifications task and Child as Transmission Agent task) and analyses of the semantic content, syntactical form, and social-emotional tenor of conversations between children and caregivers in the Caregiver-Child Conversation; network members and other researchers might also be interested in applying qualitative approaches to gauge children's certainty (e.g., their use of linguistic "hedges"), to examine the nature of the interaction between the data collector and the child, as well as other questions.

We have not provided details of these qualitative analyses here, for a combination of practical constraints and more principled reasons. As a network, the DBN is committed to allowing qualitative analyses to emerge from a long-term, iterative process of collaboration involving research team leaders, on-site staff, and additional cultural and religious experts from sites from across the network. For example, in instances where researchers will code the content of children's responses or interactions, our goal is to build coding schemes that reflect the diversity of sites and samples included in the network, rather than applying existing coding schemes out of the box. As of August 2023, this process is only in the very earliest stages.

**Data management plans.** Data from the Child Protocol and Parent Survey are collected through Qualtrics survey software. When authorized, audio and/or video recordings of the child interview are collected either using a portable recording device or using tele-conferencing software (e.g., Zoom, Microsoft Teams). These recordings are used to help transcribe responses to open-ended questions in the child interview, as well as from the Caregiver-Child Conversation.

Once datasets have been collected, data cleaning will be performed by each team. Data from participants who authorize that their de-identified responses be shared publicly will be shared on platforms for open science, including the Open Science Framework (https://osf.io/). Likewise, audio/video recordings from participants who authorize sharing these recordings will be shared with the research community on Databrary (https://www.databrary.org/).

Once all datasets from the first wave of data collection have been cleaned and de-identified, the current members of the DBN will have exclusive access to the full dataset for a finite interval of time. Following that interval, the data will then be shared publicly as described above.

**Ethical considerations and declarations.** This study has been approved by the Institutional Review Board for Socio-Behavioral research (IRB-SB) at the University of California, Riverside, under protocol #HS-21-124; and by the Institutional Review Board (Charles River IRB Office) at Boston University under protocol #4631E. Together, these IRBs cover data collection for all US samples as well as the samples in Mexico and Taiwan R.O.C. Parents/

caregivers give informed consent either in writing (on paper or via an online survey) or verbally via a live conversation with the researcher (in person or over video chat); and children give oral assent. The mode of consent varies across these samples depending on the modality of participation (online vs. in person) and on the degree to which parents/caregivers are comfortable reading written text.

All other planned samples are or will be covered by ethics boards at the home institutions of their respective research team leaders (who are independent subaward PIs). Research team leaders are also responsible for securing approval from any additional community organizations, school boards, or governmental bodies overseeing research in their field sites and samples.

**Status and timeline.** As of the initial submission of this paper (December 2022), data collection had begun in 27 (69%) of the planned samples, including five samples in which data collection has been completed (the two samples collected in Peru, the two samples collected in Uganda, and one of the four samples being collected in Indonesia); data collection for the remaining 14 (36%) of planned samples was set to begin before April 1, 2023. As of August 2023, Wave 1 data collection has begun in 97% of samples. We anticipate all samples being complete by April 1, 2024.

## Discussion

The study protocol described in this manuscript reflects roughly 18 months of intensive, iterative collaboration among members of the Developing Belief Network. The result is a mixed-methods study protocol that we believe is well-suited to describe key aspects of religious conceptual development across 39 unique cultural-religious contexts (to date).

Although we have articulated planned analyses for most elements of the study protocol, we consider the protocol as a whole to be largely exploratory in nature; our primary aims are to describe the landscape of religious cognition and development within and across the diverse populations represented in the network and to build a strong foundation for developing future, hypothesis-driven methodologies.

Open science principles are fundamental to the core mission of the DBN; our goal in this manuscript is to enable future researchers to make use of the rich datasets generated by this collaborative data collection effort and to adapt the current protocol for use in new sites and samples. We offer the following considerations to such researchers to enhance the usability and interpretability of these datasets and materials.

First, as detailed throughout this manuscript, in developing sample-specific adaptations of this protocol across the many cultural-religious and linguistic settings included in the network, we carefully balanced competing goals of tailoring the protocol to specific cultural-religious settings and ensuring some degree of network-wide standardization. All results must be interpreted in the context of this exercise. Data from any one sample requires both contextualization within the cultural-religious setting for which that protocol was adapted, and acknowledgment of the network-wide guidelines that might have constrained the process of sample-specific adaptation. By extension, comparisons *across* field sites and samples must also take into account the ways in which the protocols for each sample were both similar and different, which will impact how researchers might interpret any observed similarities and differences in results. Likewise, researchers adapting the current protocol for use in new settings might benefit from examining the existing sample-specific versions of the protocol, consulting directly with members of the DBN working in similar cultural or religious settings, or otherwise designing adaptations of the protocol that take into account the possibility of comparing to the 39 samples that are already represented in the network.

Second, future researchers will also note variability across research teams and field sites in how study populations are defined, with downstream consequences for sample-specific recruitment strategies and design decisions. For example, some research teams are recruiting participants from a population they define as "Muslim," which will result in a sample that includes a diverse range of participants practicing different forms of Islam completing the same "Muslim" adaptation of the study protocol; in contrast, other research distinguish between Sunni and Shia Muslims in their sampling, recruitment, and adaptation of the study protocol. Likewise, some research teams have designed "Christian" adaptations of the protocol, while others have developed "Protestant" vs. "Catholic" versions; and so on for other cultural-religious groups. Researchers should keep this in mind both when considering how to combine and compare data from the samples currently represented in the DBN, and when making decisions about sample-specific stimuli and recruitment plans for their own adaptations and extensions of this work.

As we described in the opening section of this paper, studies of religious cognition and behavior provide unique insights into the ways in which cognition and culture are mutually constituted, and the ways in which children come to think, behave, and experience the world around them. It is our hope that data and materials from the current study protocol, along with data from future waves of this collaborative research effort, will help shed light on this fundamental yet understudied aspect of conceptual development. Beyond this, we hope that the protocol described in this manuscript will inspire future researchers to develop and apply additional methods to complement the tasks we have developed to help deepen our understanding of this complex domain of human experience.

## Supporting information

**S1 Text.**
(DOCX)

## Acknowledgments

This work was conducted as part of a large-scale collaborative research effort by the Developing Belief Network (developingbelief.com), a network of researchers affiliated with a variety of institutions, working in diverse field sites around the world. This paper describes the study protocol designed as part of the network's first wave of data collection. Unless otherwise noted, the conceptualization, design, and implementation of studies included in this paper emerged from a collaborative process involving all members of the network during this wave.

Full information about members of the Developing Belief Network (Wave 1) is provided in the OSF repository associated with this manuscript (direct link to membership information as of August 2023: https://osf.io/f79rp).

The Developing Belief Network is led by Rebekah Richert (rebekahr@ucr.edu) and Kathleen Corriveau (kcorriv@bu.edu). As of August 2023, members include: Tamer Amin (American University of Beirut); Florencia Anggoro (College of the Holy Cross); Stav Bar-Maoz (Bar-Ilan University); Emily Burdett (Research Team PI); Yen-Ping Chang (National Tsing Hua University); Emily Chau (University of California, Berkeley); Cornelio Chay Cano (no institutional affiliation); Eva Chen (National Tsing Hua University); Meng-Ting Chen (National Tsing Hua University); Jana Chokor (American University of Beirut); Yuan-Yuan Chung (National Tsing Hua University); Lezanie Coetzee (University of the Witwatersrand); Flora Cohen (Boston University); John Coley (Northeastern University); Kathleen Corriveau (Boston University); Kelly (Yixin) Cui (Boston University); Audun Dahl (University of California,

Santa Cruz); Marissa Dalton (Boston University); Adi Danan (Bar-Ilan University); Jocelyn Dautel (Queen's University, Belfast); Helen Davis (Harvard University); Elizabeth Davis (University of California, Riverside); Adine DeLeon (Boston University); Gil Diesendruck (Bar-Ilan University); Denise Evans (Health Economics and Epidemiology Research Office; University of the Witwatersrand); Stephanie Farah (American University of Beirut); Aidan Feeney (Queen's University, Belfast); Julia Ganama (American University of Beirut); Lorena Garza (Boston University); Maliki E. Ghossainy (Boston University); Michael Gurven (University of California, Santa Barbara); Isabelle Harden (Boston University); Grace Horton (University of California, Berkeley); Hua-Chien Hsu (National Tsing Hua University); Harry Huang (National Tsing Hua University); Mei-Hui Huang (National Tsing Hua University); Benjamin Jee (Worcester State University); Jallene Jia En Chua (Nanyang Technological University); Chuan-Han Kao (National Tsing Hua University); Hannah Kramer (Queen's University, Belfast); Tamar Kushnir (Duke University); Natassa Kyriakopoulou (National and Kapodistrian University of Athens); Hea Jung Lee (University of California, Riverside); Kirsten Lesage (Boston University); Patricia Leshabana (Health Economics and Epidemiology Research Office; University of the Witwatersrand); Jacky Lin (National Tsing Hua University); Vongani Maluleke (Health Economics and Epidemiology Research Office; University of the Witwatersrand); Adva Maman (Bar-Ilan University); Ashley Marin (University of California, Riverside); Katherine McAuliffe (Boston College); Abby McLaughlin (Boston College); Carole Meyer-Rieth (University of California, Riverside); Shaun Nichols (Cornell University); Ageliki Nicolopoulou (Lehigh University); Ayse Payir (Boston University); Theodora Reiter (Boston University); Bolivar Reyes-Jaquez (University of California, Riverside); Angelique Ricard (Boston University); Rebekah Richert (University of California, Riverside); Peter Rockers (Boston University); Laura Shneidman (Pacific Lutheran University); G. S. (American University of Beirut); Irini Skopeliti (University of Patras); Mahesh Srinivasan (University of California, Berkeley); Jessa Stegall (Cornell University); Joanna Stephens (University of Nottingham); Lucy Stone (Boston University); Carrie (Jiayue) Sun (University of California, Riverside); Amanda Tarullo (Boston University); Laura Taylor (University College, Dublin); Esra Turan-Küçük (Boston University); Dilara Turut (Boston University); Kara Weisman (University of California, Riverside); Allison Williams (Boston University); Yue Yu (Nanyang Technological University); Meltem Yucel (Duke University); Xin Zhao (East China Normal University).

In addition to the named authors on this paper, the DBN wishes to acknowledge the vital administrative support of Carole Meyer-Rieth at the University of California, Riverside; and the wise advice of the DBN advisory board: Karen Adolph, Maureen Callanan, Rick Gilmore, Paul Harris, Deborah Kelemen, and Jacqueline Woolley.

## Author Contributions

**Conceptualization:** Kara Weisman, Maliki E. Ghossainy, Allison J. Williams, Ayse Payir, Kirsten A. Lesage, Bolivar Reyes-Jaquez, Tamer G. Amin, Florencia K. Anggoro, Emily R. R. Burdett, Eva E. Chen, Lezanie Coetzee, John D. Coley, Audun Dahl, Jocelyn B. Dautel, Helen Elizabeth Davis, Elizabeth L. Davis, Gil Diesendruck, Denise Evans, Aidan Feeney, Michael Gurven, Benjamin D. Jee, Hannah J. Kramer, Tamar Kushnir, Katherine McAuliffe, Abby McLaughlin, Shaun Nichols, Peter C. Rockers, Laura Shneidman, Mahesh Srinivasan, Amanda R. Tarullo, Laura K. Taylor, Yue Yu, Meltem Yucel, Xin Zhao, Kathleen H. Corriveau, Rebekah A. Richert.

**Formal analysis:** Kara Weisman, Maliki E. Ghossainy, Allison J. Williams, Ayse Payir, Kirsten A. Lesage, Bolivar Reyes-Jaquez, Kathleen H. Corriveau, Rebekah A. Richert.

**Funding acquisition:** Kathleen H. Corriveau, Rebekah A. Richert.

**Investigation:** Kara Weisman, Maliki E. Ghossainy, Allison J. Williams, Ayse Payir, Kirsten A. Lesage, Bolivar Reyes-Jaquez, Tamer G. Amin, Florencia K. Anggoro, Emily R. R. Burdett, Eva E. Chen, Lezanie Coetzee, John D. Coley, Audun Dahl, Jocelyn B. Dautel, Helen Elizabeth Davis, Elizabeth L. Davis, Gil Diesendruck, Denise Evans, Aidan Feeney, Michael Gurven, Benjamin D. Jee, Hannah J. Kramer, Tamar Kushnir, Natassa Kyriakopoulou, Katherine McAuliffe, Abby McLaughlin, Shaun Nichols, Ageliki Nicolopoulou, Peter C. Rockers, Laura Shneidman, Irini Skopeliti, Mahesh Srinivasan, Amanda R. Tarullo, Laura K. Taylor, Yue Yu, Meltem Yucel, Xin Zhao, Kathleen H. Corriveau, Rebekah A. Richert.

**Methodology:** Kara Weisman, Maliki E. Ghossainy, Allison J. Williams, Ayse Payir, Kirsten A. Lesage, Bolivar Reyes-Jaquez, Tamer G. Amin, Florencia K. Anggoro, Emily R. R. Burdett, Eva E. Chen, Lezanie Coetzee, John D. Coley, Audun Dahl, Jocelyn B. Dautel, Helen Elizabeth Davis, Elizabeth L. Davis, Gil Diesendruck, Denise Evans, Aidan Feeney, Michael Gurven, Benjamin D. Jee, Hannah J. Kramer, Tamar Kushnir, Katherine McAuliffe, Abby McLaughlin, Shaun Nichols, Peter C. Rockers, Laura Shneidman, Mahesh Srinivasan, Amanda R. Tarullo, Laura K. Taylor, Yue Yu, Meltem Yucel, Xin Zhao, Kathleen H. Corriveau, Rebekah A. Richert.

**Project administration:** Kara Weisman, Maliki E. Ghossainy, Allison J. Williams, Ayse Payir, Kirsten A. Lesage, Bolivar Reyes-Jaquez, Tamer G. Amin, Florencia K. Anggoro, Emily R. R. Burdett, Eva E. Chen, Lezanie Coetzee, John D. Coley, Audun Dahl, Jocelyn B. Dautel, Helen Elizabeth Davis, Elizabeth L. Davis, Gil Diesendruck, Denise Evans, Aidan Feeney, Michael Gurven, Benjamin D. Jee, Hannah J. Kramer, Tamar Kushnir, Katherine McAuliffe, Abby McLaughlin, Shaun Nichols, Ageliki Nicolopoulou, Peter C. Rockers, Laura Shneidman, Mahesh Srinivasan, Amanda R. Tarullo, Laura K. Taylor, Yue Yu, Meltem Yucel, Xin Zhao, Kathleen H. Corriveau, Rebekah A. Richert.

**Resources:** Kara Weisman, Maliki E. Ghossainy.

**Software:** Kara Weisman, Maliki E. Ghossainy, Kirsten A. Lesage.

**Supervision:** Kara Weisman, Maliki E. Ghossainy, Allison J. Williams, Ayse Payir, Kirsten A. Lesage, Bolivar Reyes-Jaquez, Tamer G. Amin, Florencia K. Anggoro, Emily R. R. Burdett, Eva E. Chen, Lezanie Coetzee, John D. Coley, Audun Dahl, Jocelyn B. Dautel, Helen Elizabeth Davis, Elizabeth L. Davis, Gil Diesendruck, Denise Evans, Aidan Feeney, Michael Gurven, Benjamin D. Jee, Hannah J. Kramer, Tamar Kushnir, Katherine McAuliffe, Abby McLaughlin, Shaun Nichols, Peter C. Rockers, Laura Shneidman, Mahesh Srinivasan, Amanda R. Tarullo, Laura K. Taylor, Yue Yu, Meltem Yucel, Xin Zhao, Kathleen H. Corriveau, Rebekah A. Richert.

**Validation:** Kara Weisman, Allison J. Williams, Ayse Payir.

**Visualization:** Kara Weisman.

**Writing – original draft:** Kara Weisman, Maliki E. Ghossainy.

**Writing – review & editing:** Kara Weisman, Maliki E. Ghossainy, Allison J. Williams, Ayse Payir, Kirsten A. Lesage, Florencia K. Anggoro, Emily R. R. Burdett, Eva E. Chen, Audun Dahl, Jocelyn B. Dautel, Aidan Feeney, Michael Gurven, Benjamin D. Jee, Hannah J. Kramer, Tamar Kushnir, Laura Shneidman, Mahesh Srinivasan, Meltem Yucel, Kathleen H. Corriveau, Rebekah A. Richert.

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
