## [Decision Letter · Decision Letter 0]

25 Apr 2023

PONE-D-22-32730The development and diversity of religious cognition and behavior: Protocol for Wave 1 data collection with children and parents by the Developing Belief NetworkPLOS ONE

Dear Dr. Ghossainy,

Thank you for submitting your manuscript to PLOS ONE. After careful consideration, we feel that it has merit but does not fully meet PLOS ONE’s publication criteria as it currently stands. Therefore, we invite you to submit a revised version of the manuscript that addresses the points raised during the review process. As you can see from the full reports available below, three reviewers assessed your study protocol. They appreciate the importance and timeliness of your research question, and provided positive comments regarding the study design. They also provided constructive feedback to further improve the methodological reporting and presentation of your protocol, and requested clarifications about the potential implications of your findings. Please carefully address all concerns raised.   Please submit your revised manuscript by Jun 08 2023 11:59PM. If you will need more time than this to complete your revisions, please reply to this message or contact the journal office at plosone@plos.org. Please include the following items when submitting your revised manuscript:A rebuttal letter that responds to each point raised by the academic editor and reviewer(s). You should upload this letter as a separate file labeled 'Response to Reviewers'.A marked-up copy of your manuscript that highlights changes made to the original version. You should upload this as a separate file labeled 'Revised Manuscript with Track Changes'.An unmarked version of your revised paper without tracked changes. You should upload this as a separate file labeled 'Manuscript'.If applicable, we recommend that you deposit your laboratory protocols in protocols.io to enhance the reproducibility of your results. Protocols.io assigns your protocol its own identifier (DOI) so that it can be cited independently in the future. For instructions see: https://journals.plos.org/plosone/s/submission-guidelines#loc-laboratory-protocols. Additionally, PLOS ONE offers an option for publishing peer-reviewed Lab Protocol articles, which describe protocols hosted on protocols.io. Read more information on sharing protocols at https://plos.org/protocols?utm_medium=editorial-email&utm_source=authorletters&utm_campaign=protocols.

We look forward to receiving your revised manuscript.

Kind regards,

Dario Ummarino, PhD

Senior Editor

PLOS ONE

2. Please note that in order to use the direct billing option the corresponding author must be affiliated with the chosen institute. Please either amend your manuscript to change the affiliation or corresponding author, or email us at plosone@plos.org with a request to remove this option.

3. "One of the noted authors is a group or consortium [insert name of group or team]. In addition to naming the author group, please list the individual authors and affiliations within this group in the acknowledgments section of your manuscript. Please also indicate clearly a lead author for this group along with a contact email address.

5. We note that [Figure 1] in your submission contain [map/satellite] images which may be copyrighted. All PLOS content is published under the Creative Commons Attribution License (CC BY 4.0), which means that the manuscript, images, and Supporting Information files will be freely available online, and any third party is permitted to access, download, copy, distribute, and use these materials in any way, even commercially, with proper attribution. For these reasons, we cannot publish previously copyrighted maps or satellite images created using proprietary data, such as Google software (Google Maps, Street View, and Earth). For more information, see our copyright guidelines: http://journals.plos.org/plosone/s/licenses-and-copyright.

a. You may seek permission from the original copyright holder of Figure(s) [#] to publish the content specifically under the CC BY 4.0 license. 

In the figure caption of the copyrighted figure, please include the following text: “Reprinted from [ref] under a CC BY license, with permission from [name of publisher], original copyright [original copyright year].

Natural Earth (public domain): http://www.naturalearthdata.com/.

Reviewers' comments:

Reviewer's Responses to Questions

**Comments to the Author**

1. Does the manuscript provide a valid rationale for the proposed study, with clearly identified and justified research questions?

Reviewer #1: Yes

Reviewer #2: Yes

Reviewer #3: Yes

2. Is the protocol technically sound and planned in a manner that will lead to a meaningful outcome and allow testing the stated hypotheses?

Reviewer #1: Yes

Reviewer #2: Yes

Reviewer #3: Yes

3. Is the methodology feasible and described in sufficient detail to allow the work to be replicable?

Reviewer #1: Yes

Reviewer #2: Yes

Reviewer #3: Yes

4. Have the authors described where all data underlying the findings will be made available when the study is complete?

Reviewer #1: Yes

Reviewer #2: Yes

Reviewer #3: No

5. Is the manuscript presented in an intelligible fashion and written in standard English?

Reviewer #1: Yes

Reviewer #2: Yes

Reviewer #3: Yes

6. Review Comments to the Author

You may also provide optional suggestions and comments to authors that they might find helpful in planning their study.

Reviewer #1: The authors may want to consider expanding the data collection regions to account for differences in micro-cultures. For instance, in the US, data are being collected in California and Massachusetts, two of the least religious and most socio-politically liberal regions in the US, but not in the US South, which is the most religious region and the most socio-politically conservative. I assume this would extend to other collection regions, as well.

Reviewer #2: The researchers have put forth an amendable effort to create, elaborate and document an extensive protocol addressing previously under-explored segments of developing child religiosity This vast and notable effort has well-elaborated goals presented with extensive references that support their conclusions and point to the gap in the knowledge.

It also demonstrates the meticulous planning and preparation that has gone into this research project which resulted in a well-structured report.

There are only a handful of minor items and structural issues that I would encourage the researchers to consider:

1. Citing dr. Barret in paragraph two on the religious cognitive development would seem appropriate (i.e., Barrett, J. L. (2012). Born believers: The science of children's religious belief. Simon and Schuster.)

2. Bracket missing in front of reference nr. 9 in the text

3. The segment Roadmap to current manuscript, although informative, is colloquial in form and lacks the tone that has been set beforehand and should be edited to better reflect the style of a scientific paper.

4. Subgoals for each of the project aims would do well to be equally bulleted for clarity.

5. In the Materials and Methods segment the authors state that: „Sample-specific item-selection, sample size, eligibility, inclusion and exclusion criteria, recruitment strategies, the mode of data collection, and compensation vary across samples and will be detailed in future manuscripts.”

It would be advisable to inform the reader more clearly at this point that these segments are available in this report for the US sample and that they will be available in further manuscripts for other samples. If there are some guidelines that direct an overarching strategy for determining these criteria and strategies, they could also be laid out in this segment.

6. In the same segment where exclusion criteria for the US sample are elaborated, it is stated that parental assessments of working memory can lead to exclusion. Does this translate to measured working memory limitations also? If yes, the criteria should be included. In both cases, exclusion criteria should be more transparent to allow for successful replication of the protocol.

7. The wording of the Exclusion criteria paragraph stating that it is representative of the US sample is somewhat confusing and should in style reflect the paragraph on Inclusion criteria in the Sample segment of the paper.

8. Please elaborate on the rationale behind using optional caregiver tasks.

9. It would be advisable to also report the expected/experienced time to collect the answers within each protocol and for each age bracket.

10. The report on page 22 states that research teams were encouraged to use a mix “of specific individuals (i.e., Santa Claus) and plural generics (i.e., Dragons) as well as sample specific items in the list”. Research question 2, the Religious Indicators task uses similar language where it states task properties vary and “might” focus on different qualities. For instance, the first paragraph on page 31 states: “The properties included in this task might focus on the character individually (e.g., “Here is a kid who...”) or on the character’s family (e.g., “Here is a kid whose family...”), as deemed appropriate in a given setting. Research teams have been encouraged, but not required, to select religious properties that differentiate among salient religious groups in the local context. (e.g., choosing a belief about Jesus rather than God in settings where Christians live alongside members of other monotheistic religions); the only strict requirement is that these properties should be salient aspects of religious identity and should be familiar and relevant to children, as confirmed by local informants (e.g., religious leaders, parents of young children).”

There are a few more such references that indicate an arbitrary selection of items and stimuli. If that is the case, a more extensive elaboration of the level to which such indiscriminate change to the protocol can be made, as well as a rationale on when and to what extent is the protocol malleable would be beneficial to researchers attempting to replicate these findings and use these protocols.

11. The planned analysis segments although informative lack a unifying structure and comparable style. For instance, planned analyses of the sorting task segment (page 24), in the second paragraph, starts with the sentence “We anticipate researchers analysing the data with multiple aims.“ as if to indicate what some other researchers outside of the consortium might be interested in analysing. This, however, is not what is present in other segments where the authors indicate the analysis they are or will be interested in conducting. The planned analysis segment should follow up with the elaboration of the planned analysis that are conducive to answering the research questions. Anticipation of analysis of other researchers should be left out as speculation or could be included as suggestions or recommendations. The Planned Analysis segment of the Property Attributions task (page 28) again faces the same impersonal language and inconsistency. The wording used now is “describing the most basic tasks” but it is not clear what this means. Consider using uniform structure and wording to explain the likelihood and timeline of these analyses.

12. The tone of the text switches from personal to impersonal. Especially in the planned analysis segment. For instance, in the Moral norm variation analyses the authors state they “might also examine the main effect”. In this form, this information is either superfluous or not appropriately presented. I urge the authors to avoid using “we” and shift to an impersonal style of writing as well as avoid speculating on what someone might or might not do and present what is feasible, advisable, required, or encouraged.

13. The authors in multiple segments address the need for Coding schemes (i.e., Justification task, Caregiver-Child Conversation task) but do not elaborate sufficiently on this issue for it to be informative for further research. I would propose a more detailed elaboration of the mentioned coding schemes to the scope that is appropriate for a protocol report.

14. In the Planned analyses of Religion questions segment the authors state the analysis “will be qualitative and exploratory in nature; we will not describe them here”. I would advise for the last sentence to be rephrased in order to better reflect a more formal writing style.

15. Including references that justify hypotheses put forth in the analysis section would also be advisable.

A few additional questions for the researchers:

• Protocols include questions about the COVID pandemic. Is that related to the work and goals, and how? If yes, should it be used by other researchers, and to what end?

• Should Caregiver-Child Conversation data that was collected independently be coded as such?

Finally, I would like to commend the authors for their work. Apart from these minor, and mostly style-based comments, I do not have any substantial issues with the presented work, and I wholeheartedly recommend the publication of this report.

Reviewer #3: - The manuscript describes a protocol for epic data collection for the developmental and cross-cultural study of religious thinking and behavior of children (age between 4-10) and their parents across the chosen religious traditions of present times.

- The manuscript seems to be well thought out and contains a clear and concise description of what the authors aim to do, partly why they want to do it and why in that specific way.

- As it is not an article publishing research results and conclusions, where I can consider the relationship between questions and answers, I feel uneasy about how I should actually review the presented work.

- I am used to being in the position of reviewing protocol (study designs) in the situation of considering funding, or where the feedback to authors should help to improve the design. Neither of these criteria seems to be applicable in this case. As it is certainly applaudable to make the design of the research public as a form of "pre-registration", I am not sure what is the role of peer review for such a thing.

- I generally enjoyed reading the protocol and the opportunity to think about the research problem and the complexities connected with the hardship of studying it. Even though, to my taste, the authors rely too much on a specific understanding of religion (focus on belief) and too much on cognitivist cognitive science in their theoretical and methodological perspective (behaviour is controlled by disembodied mental representations and mechanisms, which can be studied just in conversation and in discursive exercises).

- I believe, that the research can and will bring exciting new findings, I identify with the authors' claim that children's religious worldviews are an understudied phenomenon. But I do not identify with some theoretical and methodological choices. I am, however, reluctant to classify such differences as the author's problems or mistakes, which should anyhow hinder the publication or their research.

- Further notes are then brief, they point more to my experience with the text than the text itself:

- The first task for children puzzles me, and I admit there lies a root of my personal uncomfort about the design of the study. The "what is real" problem as part of the autonomous NON-4E (enacted, embodied, embedded, extended) cognition belief system is something which I connect to the beginnings of the cognitive science of religion, and I am surprised to see it in such a naked form after 20 years of existence of its empirical research program and at least a decade of various critique why especially in the case of religion, we should NOT use the methods of the laboratory cognitive psychology from the eighties.

- The study repeatedly claims it wants to study children's lived experiences, yet it seems to be more about their theological opinions than lived experiences. I believe that children's religious thinking and behaviour are even more determined by situational context than adults' thinking and behavior. I find the tasks appropriately accommodated to children in the sense of their operation, yet I feel a lot of unease with the theoretical expectations, where children are just somewhat cognitively impaired adults.

- The most missing part of the study for me is the missing theorisation of cognitive-situational frames, which could be in a developmental research context probably manifested best in the problem of "play".

- I find this large and cross-culturally broad studying children's cognition oriented on religion and fiction without considering the problem of the "magic circle", i.e. the cognitive machinery responsible for maintaining situational borders between social frames of action, a greatly missed opportunity.

- To sum up: I have a concern about the ecological validity of the findings, and I mostly fear two things. Firstly, the findings will be impaired by the unjustified reduction of lived human behaviour to a cognitivist paradigm of religious belief as out-of-context raptured mental representations and mechanisms. Secondly, the findings will reflect more on how adults guide and understand the children's representation of religion, than how children really navigate the landscape of the religious and non-religious things in their lives.

7. PLOS authors have the option to publish the peer review history of their article (what does this mean?). If published, this will include your full peer review and any attached files.

Reviewer #1: No

Reviewer #2: **Yes: **Igor Miklousic

Reviewer #3: No

---

## [Author Response · Author response to Decision Letter 0]

7 Jun 2023

We have attached a PDF with our Response to Reviewers to this re-submission.

---

## [Decision Letter · Decision Letter 1]

8 Aug 2023

PONE-D-22-32730R1The development and diversity of religious cognition and behavior: Protocol for Wave 1 data collection with children and parents by the Developing Belief NetworkPLOS ONE

Dear Dr. Weisman,

Thank you for submitting your manuscript to PLOS ONE. After careful consideration, we feel that it has merit but does not fully meet PLOS ONE’s publication criteria as it currently stands. Therefore, we invite you to submit a revised version of the manuscript that addresses the points raised during the review process.

Please address the methodological issues raised by Reviewer 4. Please submit your revised manuscript by Sep 22 2023 11:59PM. If you will need more time than this to complete your revisions, please reply to this message or contact the journal office at plosone@plos.org. Please include the following items when submitting your revised manuscript:A rebuttal letter that responds to each point raised by the academic editor and reviewer(s). You should upload this letter as a separate file labeled 'Response to Reviewers'.A marked-up copy of your manuscript that highlights changes made to the original version. You should upload this as a separate file labeled 'Revised Manuscript with Track Changes'.An unmarked version of your revised paper without tracked changes. You should upload this as a separate file labeled 'Manuscript'.If applicable, we recommend that you deposit your laboratory protocols in protocols.io to enhance the reproducibility of your results. Protocols.io assigns your protocol its own identifier (DOI) so that it can be cited independently in the future. For instructions see: https://journals.plos.org/plosone/s/submission-guidelines#loc-laboratory-protocols. Additionally, PLOS ONE offers an option for publishing peer-reviewed Lab Protocol articles, which describe protocols hosted on protocols.io. Read more information on sharing protocols at https://plos.org/protocols?utm_medium=editorial-email&utm_source=authorletters&utm_campaign=protocols.

We look forward to receiving your revised manuscript.

Kind regards,

Rosemary Frey

Academic Editor

PLOS ONE

Journal Requirements:

Reviewers' comments:

Reviewer's Responses to Questions

**Comments to the Author**

1. Does the manuscript provide a valid rationale for the proposed study, with clearly identified and justified research questions?

Reviewer #2: Yes

Reviewer #4: Yes

2. Is the protocol technically sound and planned in a manner that will lead to a meaningful outcome and allow testing the stated hypotheses?

Reviewer #2: Yes

Reviewer #4: Yes

3. Is the methodology feasible and described in sufficient detail to allow the work to be replicable?

Reviewer #2: Yes

Reviewer #4: Yes

4. Have the authors described where all data underlying the findings will be made available when the study is complete?

Reviewer #2: Yes

Reviewer #4: Yes

5. Is the manuscript presented in an intelligible fashion and written in standard English?

Reviewer #2: Yes

Reviewer #4: Yes

6. Review Comments to the Author

You may also provide optional suggestions and comments to authors that they might find helpful in planning their study.

Reviewer #2: The authors have elaborated on all the points made in the first iteration of the review process and provided clarification and amendments to the manuscript where needed and I am happy to recommend it for publication

Reviewer #4: Overview

The present manuscript details the protocol of a large scale, multi-national longitudinal study of children’s religious cognition and behavior. It is evident that the resulting data from this protocol will provide the most comprehensive data on the development of religious cognition to date, and will provide data for future researchers to examine additional questions. I found the manuscript fascinating, and I feel the need to commend the author on the time and care they took on preparing this submission. I also should acknowledge that this is my first time reviewing a study protocol. Given that this manuscript is reporting partly on the protocol of the entire study, but partly focusing on the samples from the United States, and that data collection has started (and so modifications to the actual questions or measures is not possible), I found it challenging to determine what my comments should focus on. Therefore, please consider most of my comments suggestions.

Major comments

One consistent challenge that I had in making sense of this manuscript is that in my reading the manuscript is trying to do two things at once. First, it is describing the Developing Belief Network at large and describing a basic protocol and sample description for the project as a whole. Second, it is providing a more detailed protocol for specifically the sites in the United States. This means that the authors several examples of the materials and more details on the method for the sites in the United States. For me this was confusing and kept raising questions about how several issues were going to be handled for the other samples. It did not help that the authors sometimes mention how the protocol will be adapted (e.g., selection of religious entities includes adaptation for Buddhism, even though this is not one of the religious groups being recruited in the US and details on other modification), but other times do not (e.g., sample size determination). This was odd to me as several non-US sites have started or even completed data collection, meaning that many of these decisions have been made already. I assume that the authors are trying to strike a balance in this manuscript between being comprehensive but also succinct (particularly given the current length of the manuscript), however, I think a few more details in different sections would enhance the manuscript. For example, the authors mention that each site will vary across samples, which is fine, but they can add “with at least XX children per sample” or they could state in the “Sample size (US samples)” section that a similar process will be used for the other sites.

Please include details on the translation process. In page 7, it says that this will be included in future manuscript. However, this is a critical component of the protocol and it is an aspect that I assume has already taken place (as 68% of the samples have started data collection). Therefore, I encourage the authors to include at least a basic description of their current approach.

What is the justification for the 75 participant minimum (and 100 maximum) for each sample? Why was the September 30th date selected? Is it an arbitrary date? After a certain number of months from the start of data collection? Necessary for grant reporting or for a student to graduate? Whatever the reason it is fine, but to me it seems like important information to mention in a protocol. Will similar stopping rules be used for the non-US sites? Again, I am assuming that this decision has already been made as data collection in some sites as finished, so I am surprised this is not included in the manuscript.

I have two questions regarding exclusions due to frequent interruptions. First, what percentage of question would cause a child’s data to be excluded? How will the research team ensure consistency across the sites (the two US sites at least)? Second, what about cases in which there is interference in some questions, but there is not enough to justify the exclusion of the entirety of the child’s responses. Will only the questions with interference be eliminated? Retain the response prior to the interference? These details might be worth specifying in the protocol.

I was surprised by the lack of statistical details on the protocol. What type of modeling techniques will the researchers use? How would they determine random effects structures or handle non-convergence (if multilevel models will be used)? Will they use frequentist or Bayesian models?

I appreciate the authors noting when certain analyses are exploratory. However, I found the sections on qualitative analyses to be lacking details. I read the authors’ responses to a similar comment by another reviewer, but even if the analyses are exploratory the authors can include more details on how they plan to approach the qualitative data analysis. As it currently stands, it seems like the qualitative analyses are an after-thought, which is surprising given that there are planned analyses and hypotheses regarding the qualitative data, and that the authors present this study as a mixed-methods study. Even if there is not an existing coding scheme the authors can state the type of qualitative methods they will use. Something as broad as stating that they will use thematic or content analysis (rather than other techniques like narrative analysis) would make the protocol more robust. If coding schemes are going to be developed, then how will this be done? Will they look at a certain percentage of the data to develop the coding scheme? Look at all the data? Will the coding scheme be developed inductively or deductively? I imagine it is a mix, as the hypotheses indicate that the authors already have some codes in mind. Will a particular code need to be present a certain number of times or mentioned by a certain percentage of participants for it to be included in the final scheme? The authors could also detail their coding and reliability process. Will there be multiple coders that independently code the specific percentage of the data to assess reliability? If so, what would this percentage be? How many coders? What reliability indicator would you use and what cut off point? How will you handle codes with low reliability, or low prevalence (which would impact reliability)? Would you use a form of consensus coding that would require no assessment of reliability? Many qualitative techniques differ in all of these factors so many of these decisions would be fine methodologically, but the authors should specify them a priori and include those details in the manuscript.

Minor comments

For some tasks, such as the biological/physical/psychological properties, the authors state the length of the task as whole. For other tasks, such as the religious indicators task, the authors state the length of each block. I would recommend they are consistent on which they report, and suggest they report the length of the task as a whole.

How will the authors compare the child and parent responses? Will it be done as a group? Match the children to their parent? If so, how would the authors handle siblings?

This is a minor point, but in the social essentialism task the authors include a question that involves “souls.” Given the different religious backgrounds (including children that are unaffiliated to any religion) in the entire study, will all children interpret this question similarly?

7. PLOS authors have the option to publish the peer review history of their article (what does this mean?). If published, this will include your full peer review and any attached files.

Reviewer #2: No

Reviewer #4: **Yes: **David Menendez

---

## [Author Response · Author response to Decision Letter 1]

6 Sep 2023

Please see attachment: "Response to Reviewers"

---

## [Decision Letter · Decision Letter 2]

28 Sep 2023

The development and diversity of religious cognition and behavior: Protocol for Wave 1 data collection with children and parents by the Developing Belief Network

PONE-D-22-32730R2

Dear Dr. Weisman,

We’re pleased to inform you that your manuscript has been judged scientifically suitable for publication and will be formally accepted for publication once it meets all outstanding technical requirements.

Kind regards,

Rosemary Frey

Academic Editor

PLOS ONE

Additional Editor Comments (optional):

Reviewers' comments:

Reviewer's Responses to Questions

**Comments to the Author**

1. Does the manuscript provide a valid rationale for the proposed study, with clearly identified and justified research questions?

Reviewer #2: Yes

Reviewer #4: Yes

2. Is the protocol technically sound and planned in a manner that will lead to a meaningful outcome and allow testing the stated hypotheses?

Reviewer #2: Yes

Reviewer #4: Yes

3. Is the methodology feasible and described in sufficient detail to allow the work to be replicable?

Reviewer #2: Yes

Reviewer #4: Yes

4. Have the authors described where all data underlying the findings will be made available when the study is complete?

Reviewer #2: Yes

Reviewer #4: Yes

5. Is the manuscript presented in an intelligible fashion and written in standard English?

Reviewer #2: Yes

Reviewer #4: Yes

6. Review Comments to the Author

You may also provide optional suggestions and comments to authors that they might find helpful in planning their study.

Reviewer #2: The paper is well devised, all the comments have been adequately addressed and I endorse it for publication.

Reviewer #4: The authors have addressed all my concerns. I appreciate the greater description of the qualitative analyses and the the examples of the possible paths for these analyses. The reasoning behind some of the lack of details stated in the response to was compelling to me.

7. PLOS authors have the option to publish the peer review history of their article (what does this mean?). If published, this will include your full peer review and any attached files.

Reviewer #2: No

Reviewer #4: **Yes: **David Menendez

---

## [Editor Report · Acceptance letter]

28 Feb 2024

PONE-D-22-32730R2 

PLOS ONE

Dear Dr. Weisman, 

I'm pleased to inform you that your manuscript has been deemed suitable for publication in PLOS ONE. Congratulations! Your manuscript is now being handed over to our production team.

Kind regards, 

on behalf of

Dr. Rosemary Frey 

Academic Editor

PLOS ONE